# The Challenge of Diagnosing Constitutional Mismatch Repair Deficiency Syndrome in Brain Malignancies from Young Individuals

**DOI:** 10.3390/ijms22094629

**Published:** 2021-04-28

**Authors:** Cristina Carrato, Carolina Sanz, Ana María Muñoz-Mármol, Ignacio Blanco, Marta Pineda, Jesús Del Valle, Estela Dámaso, Manel Esteller, Eva Musulen

**Affiliations:** 1Department of Pathology, Hospital Universitari Germans Trias i Pujol, 08916 Badalona, Spain; ccarrato@hotmail.com (C.C.); cs301bis@yahoo.es (C.S.); ammunoz.germanstrias@gencat.cat (A.M.M.-M.); 2Program on Clinical Genetics and Genetic Counseling, Hospital Universitari Germans Trias i Pujol, 08916 Badalona, Spain; iblanco.germanstrias@gencat.cat; 3Hereditary Cancer Program, ONCOBELL Program, Hospitalet de Llobregat, Catalan Institute of Oncology, Institut d’Investigació Biomèdica de Bellvitge (IDIBELL), 08908 L’Hospitaled de Liobregat, Spain; mpinedariu@gmail.com (M.P.); jdelvalle@iconcologia.net (J.D.V.); edamaso.riquelme@gmail.com (E.D.); 4Centro de Investigación Biomédica en Red de Cáncer (CIBERONC), 28040 Madrid, Spain; mesteller@carrerasresearch.org; 5Josep Carreras Leukaemia Research Institute (IJC), 08916 Badalona, Spain; 6Physiological Sciences Department, School of Medicine and Health Sciences, University of Barcelona (UB), 08007 Barcelona, Spain; 7Institució Catalana de Recerca i Estudis Avançats (ICREA), 08010 Barcelona, Spain; 8Department of Pathology, Hospital Universitari General de Catalunya-Grupo QuirónSalud, 08195 Sant Cugat del Vallès, Spain

**Keywords:** constitutional mismatch repair deficiency syndrome, MMR gene expression, immunohistochemistry, *MSH6* gene

## Abstract

Biallelic germline mismatch repair (MMR) gene *(MLH1, MSH2, MSH6,* and *PMS2)* mutations are an extremely rare event that causes constitutional mismatch repair deficiency (CMMRD) syndrome. CMMRD is underdiagnosed and often debuts with pediatric malignant brain tumors. A high degree of clinical awareness of the CMMRD phenotype is needed to identify new cases. Immunohistochemical (IHC) assessment of MMR protein expression and analysis of microsatellite instability (MSI) are the first tools with which to initiate the study of this syndrome in solid malignancies. MMR IHC shows a hallmark pattern with absence of staining in both neoplastic and non-neoplastic cells for the biallelic mutated gene. However, MSI often fails in brain malignancies. The aim of this report is to draw attention to the peculiar IHC profile that characterizes CMMRD syndrome and to review the difficulties in reaching an accurate diagnosis by describing the case of two siblings with biallelic *MSH6* germline mutations and brain tumors. Given the difficulties involved in early diagnosis of CMMRD we propose the use of the IHC of MMR proteins in all malignant brain tumors diagnosed in individuals younger than 25 years-old to facilitate the diagnosis of CMMRD and to select those neoplasms that will benefit from immunotherapy treatment.

## 1. Introduction

Constitutional mismatch repair deficiency syndrome (CMMRD, MIM No. 276300) is caused by biallelic germline mutations in mismatch repair (MMR) system genes [1,2]. *PMS2* is the most frequently mutated gene accounting for 60% of CMMRD cases. The remaining 40% are due to *MSH6* (20–30%), *MLH1* and *MSH2* (10–20%) pathogenic variants [3]. A unique case of homozygous *EPCAM* patient has been described [4]. It is usually seen among families of individuals with Lynch syndrome (LS) who carry monoallelic mutations in one of these genes [1,2]. CMMRD is a very rare familial syndrome with about 200 cases reported since the first description in 1999 [3]. It has an autosomal recessive pattern of inheritance and affects children and adolescents who mainly develop brain tumors, gastrointestinal neoplasms, and hematological malignancies. Other LS-related neoplasms, such as endometrial and urinary tract carcinomas, can also develop. Additionally, some features of neurofibromatosis (NF) type 1, especially café-au-lait macules (CALMs) with irregular borders and different degrees of pigmentation, may be present in the absence of *NF1/SPRED1* germline mutations [5,6]. The presence of multiple pilomatricomas [7], brain anomalies [8] and developmental vascular abnormalities has also been described [9]. Consanguinity of the parents is also frequently present [10,11]. Gastrointestinal lesions are age-dependent and include adenomas with or without high grade dysplasia as well as polyps resembling those seen in juvenile polyposis [12]. Other rare manifestations such as pediatric systemic lupus erythematosus have been described in 2.5% of CMMRD patients [13]. The pleiotropic CMMRD phenotype is not specific to the syndrome as these features overlap with other childhood tumor syndromes, in particular NF1. These features are used in a scoring system from the European C4CMMRD Consortium to raise clinical suspicion of CMMRD (Table 1) [3]:

The identification of pathogenic variants in both alleles of an MMR gene is required to confirm the diagnosis. However, the presence of *PMS2* pseudogenes [14] and variants of unknown significance (VUS) makes this difficult. To improve the previous consensus established [15] a new diagnostic criterion for CMMRD elaborated by an international panel of experts has recently been described to unmask this rare syndrome (Table 2) [4]. They established four definitive criteria to ensure diagnosis of the syndrome with strong evidence and three probable criteria with moderate evidence based on the combination of (a) the MMR germline results, (b) ancillary testing including the immunohistochemistry of the four MMR genes on non-neoplastic tissue, germline microsatellite instability [16], ex vivo MSI plus methylation tolerance [17], in vitro repair assay and NGS detection [18], low-level MSI in tissue [19,20] and (c) clinical manifestations.

In addition, guidelines for surveillance of individuals with CMMRD have been recently proposed by the European C4CMMRD Consortium [21] and the International BMMRD Consortium [22]. Even so, identification, surveillance and care of CMMRD patients continue to be a challenge.

According to recent epidemiological studies, carriers of the pathogenic variants of MMR are relatively common (up to 1 in 279 of the general population), with *PMS2* being the most frequently mutated gene [23]. In children with cancer, germline alterations have been found in up to 6% of cases, mostly affecting DNA repair genes, and the hypermutated tumors corresponded to high-grade gliomas with biallelic mutations in *MSH6* and *PMS2*. [24]. As a result, these tumors are very sensitive to immunotherapy and resistant to standard treatment with temozolomide [25].

CMMRD remains an underdiagnosed syndrome that usually debuts with malignant brain tumors in pediatric age with some clinical features that should raise suspicion of this syndrome [26] and especially in the context of high consanguinity as shown by the study of Amaya et al [11]. In this study, the prevalence of CMMRD was evaluated among patients with malignant brain tumors younger than 18 years with a high rate of consanguinity. Among the 36 high-grade gliomas, 17 (39%) showed a lack of staining for an MMR protein, and of these 82% of cases showed the characteristic pattern of CMMRD. Patients with lack of staining in both neoplastic and normal cells also had CALM, consanguinity, and a family history of cancer. In addition, the median age at diagnosis was 12.2 years, similar to that of sporadic tumors.

CMMRD tumor identification requires the use of the tests applied for LS identification, namely immunohistochemical (IHC) assessment of the four MMR proteins expression and/or analysis of microsatellite instability (MSI) [15]. In CMMRD, however, these studies show some peculiar findings, knowledge of which is mandatory to avoid underdiagnosis of this rare syndrome. Specifically, loss of MMR protein immunoreaction is not only seen in tumor cells but also in normal cells. Moreover, in CMMRD brain malignancies the standard MSI analysis fails to show instability [2,7,27,28,29,30,31,32,33,34,35,36,37,38,39,40,41,42,43,44,45,46,47,48,49,50,51,52,53,54,55,56,57,58,59,60,61,62,63,64,65,66] although the use of an extended Bethesda panel with 10 additional mononucleotide repeat markers has detected subtle frameshifts [61]. Nowadays, the use of MMR IHC is becoming increasingly widespread mainly due to universal LS screening of colorectal and endometrial tumors [67,68]. Its use will increase with the adoption of the tissue diagnostic approach, a revolutionary paradigm shift in cancer treatment and drug development [69]. Therefore, a good knowledge of the immunohistochemical pattern characteristic of CMMRD is crucial to avoid pitfalls in its interpretation, given the great diversity of lesions and neoplasms that may be the first manifestation of this syndrome. 

The aim of this report is to draw attention to the peculiar IHC profile that characterizes CMMRD syndrome, and to review the methods and difficulties in reaching a correct diagnosis. We demonstrate this by the case of two children with biallelic *MSH6* germline mutations and brain tumors, and a review of the literature.

## 2. Cases

A healthy four-years-old girl was referred to our Neurofibromatosis Clinic due to a suspected family history of NF involving her two brothers. This girl showed no NF signs at that time, but her parents were worried about her future. The older brother was born with six CALMs and was considered as a potential NF type 1 patient and, consequently, was included in a NF surveillance program. At the age of seven he had complained of headaches and vomiting and been diagnosed as having a brain tumor. The latter was totally resected and found to be a glioblastoma multiforme (GBM). The patient died at the age of eight due to a spinal cord relapse of the tumor. The younger brother was born with a single CALM but was considered as an NF type 1 patient due to his older brother’s medical history. At the age of four he developed headaches and vomiting and an NF-related pilocytic astrocytoma was suspected in an MRI study. A conservative management was initially recommended. However, headaches were persistent and an MRI performed two months later showed a growing tumor. A surgical resection was carried out but, unfortunately, a complicated post-operative course led to death two weeks after surgery. The pathological diagnosis was malignant astrocytoma. There was no additional family history of NF. The parents were healthy and non-consanguineous. The paternal and maternal grand-mothers had been diagnosed with cancer, colorectal cancer at the age of 55 and endometrial cancer at 54, respectively. The family pedigree is shown in Figure 1.

We challenged the initial diagnosis of NF in this family due to the absence of previous family history of the disease and the paucity of NF manifestations, although a gonadal mosaicism in one of the progenitors could not be ruled out. 

### 2.1. Immunohistochemical Analysis of MMR Proteins 

Both brain tumors showed positive nuclear immunostaining in neoplastic cells as well as in adjacent normal brain tissue with antibodies against MLH1, PMS2, and MSH2 proteins. In contrast, absence of MSH6 expression was observed in both tumor cells and the surrounding normal tissue (Figure 2). An external positive control tissue from another individual was also studied to assess the quality of the immunostaining in each case. 

### 2.2. MSI Analysis

Analysis of MSI showed that none of the brain tumors were MSI-H. In the GBM case all markers showed an apparently stable profile (Figure 3), whereas in the malignant astrocytoma case only the NR27 marker showed a broader profile. This may represent a germline variant as this specific marker is known to be quasi-monomorphic [70]. 

### 2.3. Germline Mutation Analysis

Sequence analysis of the *MSH6* gene in father’s blood sample displayed a heterozygous single nucleotide duplication in exon 5 (c.3261dupC) leading to a frameshift and premature termination (p.Phe1088Leufs*5). The same analysis in the mother’s sample showed a heterozygous single nucleotide duplication in exon 4 (c.762dupT) leading to a nonsense mutation, p (Glu255 *). Both variants were classified as pathogenic (class 5). Analysis of the two brothers’ tumor brain tissues confirmed that both were compound heterozygous *MSH6* c.3261dupC and c.762dupT mutation carriers. Our patient (III.3 Figure 1), the four-year-old girl, was heterozygous for the c.762dupT *MSH6* gene mutation. This result ruled out the diagnosis of CMMRD syndrome and she was diagnosed with LS. 

### 2.4. Somatic Mutational Analysis

Acquired polymerase proofreading defects are characteristic of CMMRD brain tumors, leading to high mutational burden and associated with response to immune checkpoint inhibitors [25,71,72]. Therefore, the analysis of *POLE* and *POLD1* genes in the brain tumor DNA sample from patient III-2 revealed the presence of four variants in *POLD1* (Table 3). Of note, one of these variants, p.Glu318Lys affected the catalytic residue of *POLD1* and is classified as pathogenic. Unfortunately, the mutational analysis was performed after the patient’s death and could not be used to guide an appropriated treatment. 

## 3. Brain Tumors in CMMRD Syndrome

In order to compare our findings with those previously reported, the results of MMR protein expression using IHC and MSI analysis from a total of 118 reported brain tumors in patients with CMMRD syndrome have been reviewed and are specified in Table 4 [2,7,27,28,29,30,31,32,33,34,35,36,37,38,39,40,41,42,43,44,45,46,47,48,49,50,51,52,53,54,55,56,57,58,59,60,61,62,63,64,65,66].

Of the 118 patients in the series, the gender was specified in 77, 43 (56%) males and 34 (44%) females. 

Regarding the distribution of tumors by age, 72 (61%) appeared in the first decade of life, 26 (22%) between 11 and 15 years, 6 (5%) between 16 and 20, 7 (6%) between 21 and 24 and 6 (5%) over 25 years. 

The most frequently altered gene was *PMS2* with 68 (63%) cases, followed by *MSH6* (27, 25%), *MLH1* (10, 9%) and *MSH2* (3, 3%). In 10 cases there is no data available on the altered gene.

MMR IHC was not performed in 59 tumors. Among the 59 cases analyzed, loss of PMS2 expression was the most frequent finding with 31 (52.5%), *MSH6* was negative in 14 (23.7%), MLH1 and PMS2 negative in seven (11.8%), MLH1 negative in two (3%), MSH2 and *MSH6* negative in one (1.5%). PMS2 was positive in one (1.5%) (case 42) and MSH6 positive in three (5%) (cases 7, 96, and 104). The sensitivity of MMR IHC for identifying patients affected by CMMRD was 93.2%, as it showed loss of expression in at least one protein in 55 of the 59 brain neoplasms analyzed. In 54 cases it was possible to compare the result of germline analysis with MMR IHC and a good agreement was observed in 47 (87%). The remainder seven (13%) with non-concordant results were as follow: three mutated *MHS6* with positive *MSH6* (cases 7, 96, and 104), two mutated *PMS2* with negative MLH1 and PMS2 (cases 85 and 86), one mutated *MSH6* with negative *MSH2* and *MSH6* (case 97) and one mutated *PMS2* with positive PMS2 (case 42).

MSI analysis was performed in 56 (47%) tumors, but in 16 the results were not specified [48]. The results of the 40 tumors with available data showed that MSI-H was the most frequent finding with 19 (47.5%) followed by MSS with 16 (40%) and MSI-L with five (12.5%).

## 4. Discussion

The identification of individuals with CMMRD syndrome remains a challenge and a high degree of clinical awareness of its features and diagnostic pitfalls is required.

The cases presented here illustrate how a multidisciplinary approach, with the participation of pathologists, dermatologists, geneticists, pediatricians, gastroenterologists, and neurooncologists, is needed to ensure an early diagnosis of CMMRD in brain tumor patients as well as to provide the best treatment. MMR deficient high-grade gliomas are resistant to standard treatments with combination of temozolomide and radiation but sensitive to immunotherapy. In addition, the overall survival of patients with CMMRD and malignant brain tumors is poor and worse than seen in the same population with no predisposition to hereditary cancer [26]. For example, today, according to C4CMMRD criteria, [3] the clinical suspicion of CMMRD should have been raised in the older sibling, who with more than 5 CALMs met the suspicion of NF1 at the time of diagnosis of GBM. The clinical presentation of the second sibling would be even more obvious by having a suspected NF1 skin lesion, the background of the sibling with a glioma, and in both parental lines second-degree relatives with LS-associated carcinomas before the age of 60 years. At that point, reaching a score superior to three points in the C4CMMRD criteria, further analysis should be done to confirm a diagnosis of CMMRD. The finding of a somatic pathogenic variant in *POLD1* in the high-grade glioma of this patient suggested the hypermutant nature of the tumor that characterizes brain malignancies developed in CMMRD. 

IHC study of MMR is one of the ancillary tests included, since in CMMRD at least one protein must be totally negative in normal tissue. For that reason, any unusual immunostaining pattern of MMR protein in neoplastic and normal cell should be a clue to guide the germ-line analysis. Herein, in the two new cases of CMMRD by biallelic *MSH6* mutations with brain neoplasm, the IHC study of MMR proteins was crucial to arrive at the correct molecular diagnosis. The characteristic pattern of expression seen in CMMRD, namely lack of nuclear expression in both the tumor and the normal cells with MSH6 protein, was crucial. To avoid the pitfall of interpreting the lack of staining in all cells as a failure of the technique, it is important to include an external positive control from another individual in each determination. MMR IHC is a useful tool for identifying CMMRD patients with brain neoplasms with a sensitivity of 93% based on the cases reviewed in Table 4, showing loss of expression in at least one protein in 55 of the 59 tumors analyzed. However, unusual results of MMR protein expression have also been found. Tumors from patients with a homozygous *MLH1* mutation displayed negative MLH1 staining only in neoplastic cells, with loss of PMS2 in both neoplastic and normal cells [60]. Baas et al described a spinal cord astrocytoma from a biallelic *PMS2* mutation carrier with loss of MLH1 in tumor cells, probably as a result of somatic alteration, with lack of PMS2 staining in tumor and normal tissue [8]. In addition, MSH6 and PMS2 could retain expression in tumors in the case of biallelic germline missense mutations because of residual protein expression [29,45,58,61]. This is observed in four tumors in the series reviewed (cases 7, 42, 96, and 104) and could explain the lower correlation between the loss of protein expression with the mutated gene in brain tumors compared to what occurs in other types of neoplasms such as colon and endometrial carcinoma. Another widely available assay to guide germline diagnosis of CMMRD is tumor MSI analysis, but CMMRD brain tumors were often MSS or MSI-L. The two brain neoplasms herein reported were MSS and the other MSI-L due to minor alterations with the mononucleotide NR27 marker. NR27 is one of the five markers of the quasi-monomorphic panel (NR21, NR27, NR24, BAT25 and BAT26) described by Buhard et al [70] and different from those validated in colorectal cancer due to its better specificity, which allows the avoidance of normal cells as a control [73]. However, this panel was not as efficient in detecting MSI in CMMRD tumors [52]. In contrast, dinucleotide repeat markers have been shown to be most useful in CMMRD-associated brain tumors, although they are insensitive to *MSH6* deficiency. In fact, this was well illustrated by Nguyen et al., analyzing different tumors of CMMRD patients with mononucleotide and dinucleotide repeat markers [60]. One sarcoma and two brain tumors were MSI-H only in dinucleotide repeat markers while one colorectal cancer was unstable in all types of marker. Of the 19 MSI-H brain tumors, the markers used in the analysis were detailed in 10 cases and in nine of these the markers were dinucleotide repeats [26,38,41,52,53]. Bakry et al., described 16 brain tumors and only one was MSI-H but the markers used were not specified [56]. Another way to enhance the efficiency of MSI analysis has been to extend the number of non-coding mononucleotide repeat markers. Using this approach with 10 markers, Maletzki et al demonstrated MSI-H in 3/3 brain tumors analyzed (Table 4) [61]. Even so, MSI in the identification of CMMRD patients with brain tumors showed a low sensitivity of 47.5%, which was not enough to unmask half of the patients affected by the syndrome. Another possible option that would allow the identification of CMMRD in brain tumors would be sequencing to determine MSI, mutation burden and signatures, and other alterations characteristic of these tumors, such as *POLE* or *POLD1* variants. In fact, a recent report has revealed novel signatures that are uniquely attributed to mismatch repair deficiency by using exome- and genome-wide microsatellite instability analysis [74]. In this study different microsatellite (MS)-mutated loci, lack of recurrently mutated MS-loci and lack of long MS-indels have been identified as the main differences between childhood and adult MMR-deficient cancers. These differences may explain why conventional electrophoretic MSI assays, based on the detection of indels of >3 bases in a small number of MS analyzed, were unable to detect MSI in pediatric tumors. Germline analysis confirms the suspicion of CMMRD only if pathogenic variants in both alleles of an MMR gene are found. Unfortunately, this approach is not always definitive due to the detection of VUS and the presence of pseudogenes of *PMS2* [14], the most frequent gene affected in CMMRD patients. In CMMRD, MSI may be detected in DNA from all normal cells, which is a hallmark of the syndrome. Following this principle, different techniques have been developed to detect low-frequency microsatellite length variants. Ingham et al [16] described the germline MSI (gMSI), a PCR-based assay with the DNA from the peripheral blood leukocytes (PBLs) detecting dinucleotide repeats that are not sensitive to loss of MSH6 activity. Bodo et al. described an ex vivo MSI (evMSI) using mononucleotide repeats but requiring a long process of culturing of lymphoid precursor cells as well as analysis of alkylation tolerance [17]. Recently, Gallon et al [19] perfected the strategy with the use of short (7–12 bp) mononucleotide repeats to detect low-frequency microsatellite length variants in PBLs. Their smMIP and sequencing-based assay can also be used as a screening test as it is an inexpensive test suitable for high throughput detection of at-risk populations. A promising new approach to detecting highly sensitive MSI, hsMSI, based on a panel of unstable monomorphic markers allows the identification of gMSI in normal cells in PBLs and minimally invasive samples showing robust results in the diagnosis of CMMRD [20]. Both next generation sequencing-based approaches do not have the limitations of the previous tests and provide an accurate test for CMMRD regardless of the affected gene.

In view of all the aforesaid data and in accordance with the recent consensus of diagnostic criteria [4], we propose the use of MMR IHC in malignant brain tumors diagnosed at age younger than 25 years-old to identify those with CMMRD that may have gone undetected due to few accompanying key manifestations or lack of family history of LS spectrum neoplasms. Cases reviewed of the literature in this study showed that the majority of brain tumors occurred before the age of 16 years (98/117, 84%) and 95% of cases occurred up to the age of 25 years (111/117). Only six (5%) patients were older than 25 years (27, 32, 34 (×2), 35, 40). However, given the controversy in considering “MMR-deficiency” in adult brain tumors as a predictor of treatment [75,76,77], and because these neoplasms are not included in the LS tumor spectrum, we do not believe it is appropriate to consider universal screening by IHC in brain tumors in adults older than 25 years.

Of all the available ancillary tests, we believe that MMR IHC is the most suitable for use as a screening method because it is a simple, fast, efficient, and economical technique, fully implemented in pathology departments, which allows the suspicion of CMMRD with a high sensitivity (93.2%) when MMR proteins are not expressed in tumor cells or normal adjacent cells. It is crucial for the pathologist to know the expression pattern that characterizes CMMRD to avoid misinterpretation.

To integrate MMR IHC screening into the established consensus clinical guidelines [3,4], we believe that the best strategy to follow would be to screen for CMMRD using both the MMR IHC in brain malignancy neoplasms in younger than 25 years of age and the C4CMMRD clinical scoring system [3] to select patients for germline genetic testing, those with an MMR-deficient tumor or with a score of 3 or more points; and that the final diagnosis would be made according to the recent guidelines of the International CMMRD Consortium [4]. Thus, patients excluded from MMR IHC screening due to presenting with brain tumor over the age of 25 years would be identified as CMMRD using the existing clinical scoring system of the C4CMMRD consortium based on the set of clinical features that make up their phenotype. This is true for all six patients over 25 years of age in the series reviewed (Table 4). In addition, those brain tumors retaining PMS2 or MSH6 expression were also identified applying the clinical scoring system (Table 5).

At the same time, it will ensure the selection of an effective treatment, as MSI-H tumor cells are resistant to O6-methylating agents such as temozolomide [78], but may be good candidates for immunotherapy treatments [79,80]. 

## 5. Material and Methods

### 5.1. Immunohistochemical Analysis of MMR Proteins 

Formalin-fixed, paraffin-embedded tissue sections representative of both brain tumors were studied using standard IHC techniques. The mouse primary antibodies used were the following: anti-hMLH1 (clone G168-15, PharMingen, San Diego, CA, USA), anti-hMSH2 (clone G219-1129, PharMingen, San Diego, CA, USA), anti-hMSH6 (clone 44, Transduction Laboratories, San Diego, CA, USA), and anti-PMS2 (clone A16-4, PharMingen, San Diego, CA, USA). Nuclear immunoreaction in lymphocytes, normal brain or endothelial cells served as positive control. A separate tissue section was used as an additional positive control since internal positive control staining is not observed in CMMRD tissues.

### 5.2. MSI Analysis

Tumor genomic DNA from the two brain tumor patients was obtained from paraffin-embedded tissue. Briefly, 5 mm hematoxylin-stained paraffin sections were deparaffinized with xylene and rehydrated with ethanol. Tissue was manually macro-dissected for tumor cell enrichment. Finally, DNA was isolated using QIAamp DNAmicrokit (Qiagen GmbH, Hilden, Germany). Five mononucleotide microsatellites (NR21, NR24, NR27, BAT25 and BAT26) were studied using the primers and conditions previously described by Buhard et al. [70]. Fragments were analyzed on an ABI PRISM 3100 Genetic Analyzer (Applied Biosystems/Thermo Fisher Scientific, Waltham, MA, USA).

### 5.3. Germline MSH6 Mutational Analysis

*MSH6* germline point mutation analysis was performed on genomic DNA isolated from peripheral blood leucocytes from patients’ parents using PCR and direct sequencing of the whole coding sequence and intron–exon boundaries. The identified *MSH6* mutations were subsequently analyzed in brain tumor DNA from their affected sons and blood sample from the healthy daughter. Variant classification was determined according to InSIGHT classification guidelines [81].

### 5.4. Targeted Next Generation Sequencing Analysis

The brain tumor DNA of patient III-2 was analyzed using a targeted Next Generation Sequencing custom panel, previously described in [82], which included the coding region of *POLE* and *POLD1* genes, among other genes (*APC, AXIN2, BMPR1A; BUB1, BUB1B, BUB3, CDH1, CHEK2, ENG, EPCAM, EXO1, FAN1, MLH1, MLH3, MSH2, MSH3, MSH6, MUTYH, PMS1, PMS2, POLD1, POLE, PTEN, SMAD4, STK11* and *TP53*) and somatic hotspot mutations (design size: 319 Kb). Tumor from patient III-1 did not fulfill quality requirements to be analyzed. Briefly, capture of the target regions was performed using HaloPlex Target Enrichment kit 1-500 kb (Agilent Technologies, Santa Clara, CA, USA), according to the HaloPlex Target Enrichment System-Fast Protocol Version B. Library concentrations were normalized to 0.44 nM. Pooled libraries were sequenced in a MiSeq (Illumina, San Diego, CA, USA) with paired-end 250 bp reads plus an 8-base index read, using MiSeq Reagent Kit v3. Agilent SureCall application was used to trim, align and call variants. Variant filtering was performed based on Phred quality ≥30, alternative allele ratio ≥0.20, read depth ≥30x in the analyzed FFPE sample. Identified variants were then filtered against common single-nucleotide polymorphisms (MAF>1 according to the *Exome* Aggregation Consortium (*ExAC*) and the NHLBI GO Exome Sequencing Project (ESP) data). Pathogenicity assessment of the identified variants was aided by Alamut software v2.9.0. (Sophia Genetics SA, Saint Sulpice, Switzerland). The size of the panel used prevented the analysis of tumor mutational burden or mutational signatures.

## 6. Conclusions

Recognizing the lack of MMR protein expression in both tumor cells and normal tissue as a pattern of CMMRD-related IHC is of crucial importance. The IHC approach is particularly relevant in brain malignancies in which MSI analysis may fail to reveal an MSI-H pattern using a panel of mononucleotide repeat markers. Therefore, we propose the use of MMR IHC in malignant brain tumors diagnosed below 25 years of age to unmask those suffering from CMMRD that may have gone undetected due to the few key manifestations or the lack of family history of LS spectrum neoplasms.

## Figures and Tables

**Figure 1 ijms-22-04629-f001:**
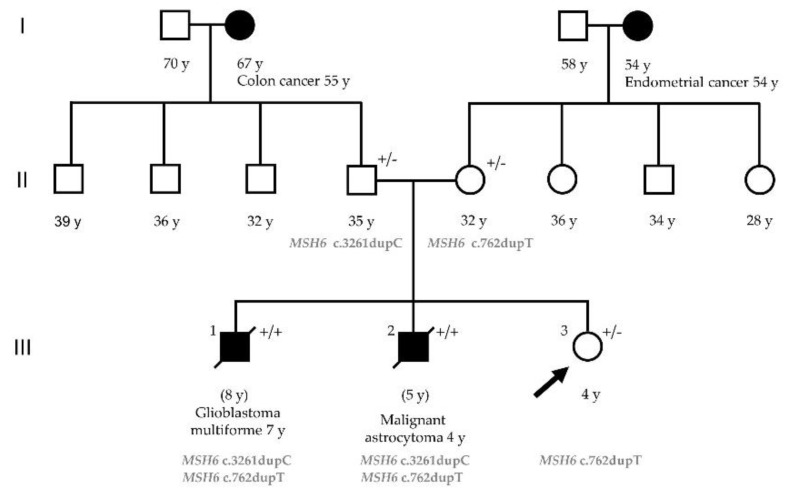
Pedigree of the family with biallelic *MSH6* mutations. The arrow indicates the proband. Filled symbols indicate affected subject; open symbols unaffected subjects. The age at tumor onset is shown with the tumor type. +/− = *MSH6* monoallelic mutation carrier (Lynch syndrome); +/+ = *MSH6* biallelic mutation carrier (CMMRD).

**Figure 2 ijms-22-04629-f002:**
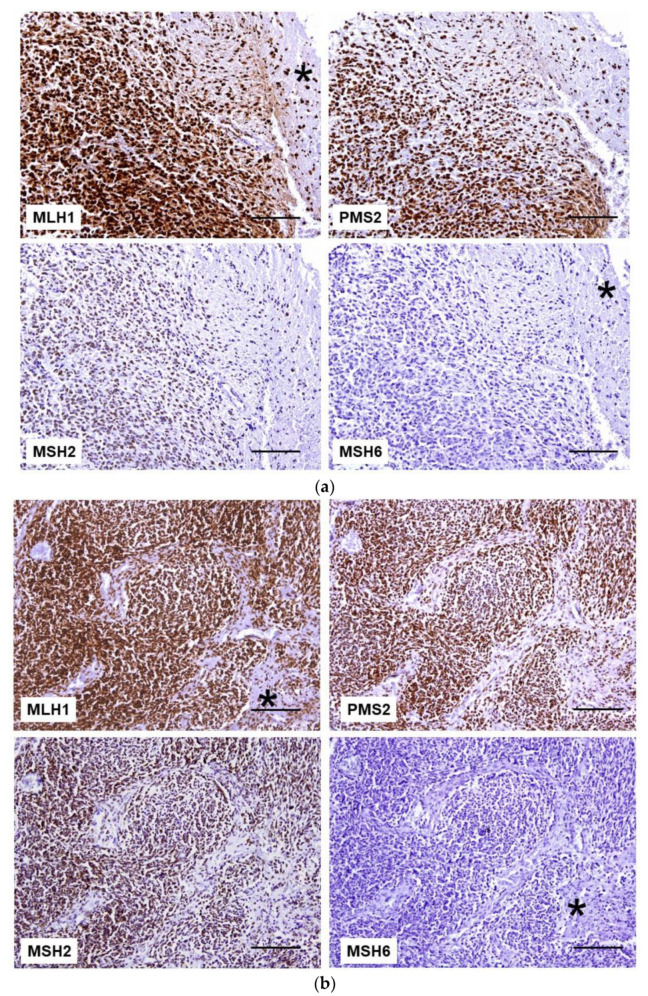
MMR protein IHC in brain tumors. (**a**) Glioblastoma cells and normal brain tissue (*) retained the expression of MLH1, PMS2, and MSH2 proteins. MSH6 staining is lost in both tumor cells and normal tissue (*). (**b**) In malignant astrocytoma, neoplastic, stromal, and endothelial cells are immunoreactive to anti-MLH1 (*), anti-PMS2, and anti-MSH2 antibodies. In contrast, the lack of *MSH6* staining is observed in all cells (*). Scale bar: 100 µm.

**Figure 3 ijms-22-04629-f003:**
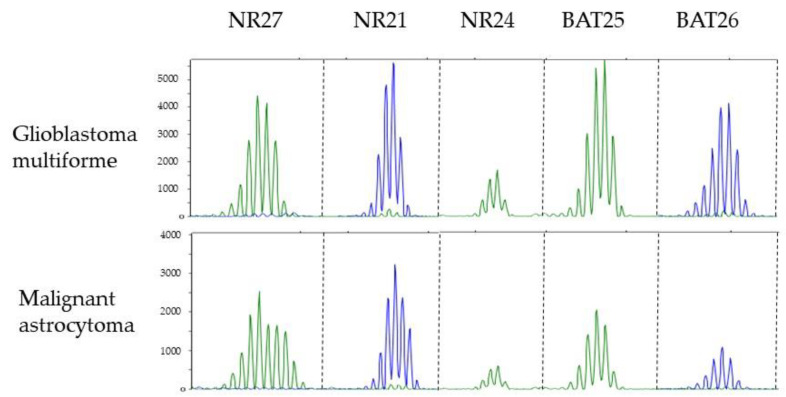
Electropherograms showing microsatellite profiles of five mononucleotide repeats markers (NR21, NR24, NR27, BAT25, and BAT26) in each tumor. A broader spectrum is observed in marker NR27 in the malignant astrocytoma.

**Table 1 ijms-22-04629-t001:** Scoring system to determine germline testing in eligibility for CMMRD.

Indication for CMMRD Testing in a Cancer Patient	≥3 Points
Malignancies/premalignancies: one is mandatory; if more than one is present in the patient, add the points	
Carcinoma from the LS spectrum * at age <25 years	3 points
Multiple bowel adenomas at age <25 years and absence of APC/MUTYH mutation(s) or a single high-grade dysplasia adenoma at age <25 years	3 points
WHO grade III or IV glioma at age <25 years	2 points
NHL of T-cell lineage or sPNET at age <18 years	2 points
Any malignancy at age <18 years	1 point
**Additional features: optional; if more than one of the following is present, add the points**	
Clinical sign of NF1 and/or ≥2 hyperpigmented and/or hypopigmented skin alterations Ø > 1 cm in the patient	2 points
Diagnosis of LS in a first-degree or second-degree relative	2 points
Carcinoma from LS spectrum * before the age of 60 in first-degree, second-degree, and third-degree relative	1 point
A sibling with carcinoma from the LS spectrum *, high-grade glioma, sPNET or NHL	2 points
A sibling with any type of childhood malignancy	1 point
Multiple pilomatricomas in the patient	2 points
One pilomatricoma in the patient	1 point
Agenesis of the corpus callosum or non-therapy-induced cavernoma in the patient	1 point
Consanguineous parents	1 point
Deficiency/reduced levels of IgG2/4 and/or IgA	1 point

* Colorectal, endometrial, small bowel, ureter, renal pelvis, biliary tract, stomach, bladder carcinoma. Abbreviations: CMMRD, constitutional mismatch repair deficiency; LS, Lynch syndrome; NHL, non-Hodgkin’s lymphomas; sPNET, supratentorial primitive neuroectodermal tumors.

**Table 2 ijms-22-04629-t002:** CMMRD diagnostic criteria.

Criterion		Germline result * *PMS2, MSH6, MSH2, MLH1*	Positive Ancillary Testing †	Clinical Phenotype
Definitive diagnosis (strong evidence of CMMRD)	1	Biallelic pathogenic variants (P/P) *, confirmed in trans. ‡	Not required, unless unaffected >25 years then one required. †	Not required if under age 25 (if no malignancy over age 25, ancillary testing required).
	2	Biallelic P/LP or LP/LP * variants, confirmed in trans ‡	One required† for hallmark CMMRD. Two required† for C4CMMRD criteria. **	Hallmark CMMRD cancer diagnosis or C4CMMRD criteria of 3 points ** (then two ancillary tests required).
	3	Heterozygous P or LP variant (±VUS * or likely benign variants).	One required †.	Hallmark CMMRD cancer diagnosis.
	4	No P or LP MMR variants (including VUS/ VUS). †† *Or* no testing available (i.e., deceased proband).	Two required †.	Hallmark CMMRD cancer diagnosis.
Likely diagnosis (moderate evidence of CMMRD)	5	Biallelic P/LP * or LP/LP variants, confirmed in trans.§	Not required.	C4CMMRD criteria of 3 points. **
	6	No P or LP MMR variants (including VUS/ VUS). †† *Or* no testing available (i.e., deceased proband).	One required. †	Hallmark CMMRD cancer diagnosis.
	7	Heterozygous P or LP variant or no testing available (i.e., deceased proband).	Two required. †	C4CMMRD criteria of 3 points. ** Individuals aged <18 with NF1 features (i.e., no malignancy or polyposis history). Malignancy under age 30

* Biallelic–impacts same gene on both parental alleles (i.e., PMS2/PMS2); P, pathogenic (ACMG C5); LP, likely pathogenic (ACMG C4); VUS (ACMG C3). Multigene panel testing is recommended to investigate overlapping conditions. Consider phenotype of individual to rule out overlapping syndromes. All families should be assessed in a specialized center for diagnosis. † Ancillary testing does not include tumor mutation burden and signature at this time. Functional testing should be published with proven high sensitivity and specificity performed in an accredited (e.g., CAP-inspected) laboratory authorized to give a clinically usable report. If discrepancy occurs among tests, multiple ancillary tests should be used to reach a more conclusive decision. ‡ In trans variants can be proven by testing parents, offspring or other relatives. If unavailable to confirm variants in trans, individual should fulfil criterion 3. § If unavailable to confirm variants in trans, individual should fulfil criterion 6. Hallmark CMMRD cancer: glioma or CNS embryonal tumors 25 years, hematological cancer (excluding Hodgkin’s lymphoma) <18 years, GI adenocarcinoma <25 years, or >10 adenomatous GI polyps <18 years (after ruling out polyposis conditions). ** C4CMMRD criteria outlined in Table 1. †† Consanguinity further supports a diagnosis of CMMRD due to a homozygous MMR gene mutation that is unidentifiable. ACMG, American College of Medical Genetics; CAP, College of American Pathologists; C4CMMRD, European Consortium Care for CMMRD; CMMRD, constitutional mismatch repair deficiency syndrome; CNS, Central Nervous System; GI, gastrointestinal; MMR, mismatch repair; NF1, neurofibromatosis type 1.

**Table 3 ijms-22-04629-t003:** *POLD1* variants found in the analysis of the brain tumor of individual III.2. In silico predictions based on Alamut software.

Variant Calling			Coverage		Rs ID	MAF
Gene (transcript)	cDNA change	Predicted protein change	Allelic frequency	Read depth		(ExAC/ESP)
*POLD1* (NM_001256849.1)	c.952G>A	p.Glu318Lys	0.264	772	rs775232133	0.000010/NR
	c.1096G>A	p.Ala366Thr	0.235	267	–	NR/NR
	c.2788G>A	p.Ala930Thr	0.482	410	rs144111108	0.00010/0.00008
	c.3157C>T	p.Arg1053Cys	0.62	91	rs779208942	0.00004/NR
	**In Silico Predictions**					**Protein Domain**
	**Protein Function**					
Gene (transcript)	Splicing	SIFT (score)	Mutation Taster (p-value)	Polyphen2/HumDiv (score)	Polyphen2/HumVar (score)	
*POLD1* (NM_001256849.1)	No changes	Deleterious (0)	Disease causing (1)	Probably damaging (1.000)	Probably damaging (0.998)	Exonuclease(catalytic residue)
	Unconclusive	Deleterious (0.05)	Disease causing (0.956)	Benign (0.087)	Benign (0.271)	Exonuclease
	No changes	Tolerated (0.13)	Disease causing (1)	Probably damaging (1.000)	Probably damaging (0.998)	–
	No changes	Deleterious (0)	Disease causing (1)	Probably damaging (0.999)	Probably damaging (0.973)	–

Abbreviations: NR, not reported.

**Table 4 ijms-22-04629-t004:** Clinicopathological and molecular features of brain tumors in CMMRD syndrome.

Chronological Case Order	Reference (Number)	Case ID	Age at Diagnosis (Years)	Gender	Histology	Altered Gene	IHC	MSI (Marker)
1	Wang [2]	1	18	M	Medulloblastoma	*MLH1*	NA	* MSI-H
2		2	7	F	Medulloblastoma	*MLH1*	NA	NA
3	De Rosa [27]	1	14	F	Oligodendroglioma GIII	*PMS2*	NA	NA
4		2	13	F	Neuroblastoma	*PMS2*	NA	NA
5	Vikky [28]	1	4	F	Glioma	*MLH1*	NA	NA
6	Bougeard [29]	1	3	M	GBM	*MSH2*	NA	MSS
7	Menko [30]	1	10	M	Oligodendroglioma	*MSH6*	MSH6+	MSS
8	De Vos 2004 [14]	1	8	M	PNET	*PMS2*	NA	NA
9		2	14	F	PNET	*PMS2*	NA	NA
10	De Vos 2006 [31]	1	8	NS	PNET	*PMS2*	NA	NA
11		2	4	NS	PNET	*PMS2*	NA	NA
12		3	15	NS	Glioma	*PMS2*	NA	NA
13		4	6	NS	Astrocytoma	*PMS2*	NA	NA
14		5	7	NS	GBM	*PMS2*	NA	NA
15		6	2	NS	Giant cell glioblastoma	*PMS2*	NA	NA
16	Agostini [32]	1	18	M	Giant cell glioblastoma	*PMS2*	PMS2-	MSS
17	Ostergaard [33]	1	9	M	Astrocytoma GIII	*MSH6*	MSH6-	NA
18		2	2	F	GBM of spinal cord	*MSH6*	MSH6-	NA
19	Hegde [34]	1	8	F	GBM	*MSH6*	NA	MSI-H (D2S123, D17S250, BAT25, D18S35, TP53-DI, D1S283, TP53-Penta, FGA, NR-21, NR-22, NR-24)
20	Durno [35]	1	14	F	Anaplastic astrocytoma	*MLH1*	NA	NA
21	Krüger [36]	1	9	M	GBM	*PMS2*	NA	NA
22		2	6	F	GBM	*PMS2*	NA	NA
23		3	9	M	GBM	*PMS2*	NA	NA
24	Gururangan [37]	1	19	M	Astrocytoma GIII	*PMS2*	PMS2-	MSS
25		2	11	M	Anaplastic oligodendroglioma	*PMS2*	NA	NA
26	Auclair [38]	1	7	F	GBM	*MSH6*	* MSH6–	* MSI-L
27		2	19	F	Oligodendroglioma	*PMS2*	* PMS2–	* MSI-H
28	Poley [39]	1	4	M	GBM	*MLH1*	MLH1–	MSS
29		2	6	M	Oligodendroglioma	*MSH6*	MSH6–	MSI-L
30		3	8	M	Medulloblastoma	*MSH6*	MSH6–	MSI-L
31	Scott [40]	1	7	F	Medulloblastoma	*MSH6*	MSH6–	NA
32	Kratz [41]	1	9	F	PNET	*PMS2*	NA	NA
33	Etzler [42]	1	6	NS	Medulloblastoma	*MSH6*	NA	NA
34		2	9	NS	GBM	*MSH6*	MSH6–	MSS
35		3	10	NS	GBM	*PMS2*	PMS2–	MSS
36	Senter [43]	1	23	NS	Brain tumor	*PMS2*	NA	NA
37		2	35	NS	Glioma	*PMS2*	NA	NA
38		3	7	NS	Medulloblastoma	*PMS2*	NA	NA
39	Tan [44]	1	8	M	GBM	*PMS2*	NA	NA
40	Toledano [45]	1	NS	M	Anaplastic astrocytoma GIII	*MSH2*	NA	NA
41		2	13	M	Anaplastic astrocytoma GII	*MSH2*	NA	NA
42	Sjursen [46]	1	10	F	Giant cell glioblastoma	*PMS2*	PMS2+	MSI-H (BAT25, BAT26,BAT40,D2S123,D5S107, D5S346, D5S406,D13S153, D17S250)
43	Giunti [47]	1	10	M	GBM	*PMS2*	NA	NA
44		2	4	F	Brain tumor	*PMS2*	NA	NA
45	Roy [48]	1	8	F	Medulloblastoma	*PMS2*	NA	NA
46	Herkert [49]	1	2	M	Angiosarcoma cerebral	*PMS2*	PMS2-	MSI-H (BAT25, BAT26, D2S123, D5S346, D17S250)
47		2	34	M	GBM	*PMS2*	NA	NA
48		3	9	F	Anaplastic ganglioma	*PMS2*	PMS2-	MSI-H (BAT25, BAT26, D2S123, D5S346, D17S250)
49	Ilencikova [50]	1	11	F	Fibrillar astrocitoma	*MSH6*	NA	NA
50		1	12	M	Anaplastic astrocytoma GIII	*MSH6*	MSH6-	MSI-L (BAT26)
51		2	10	M	GBM	*MSH6*	NA	NA
52	Leenen [51]	1	4	F	PNET	*PMS2*	PMS2-	MSS
53		2	7	M	Glioblastoma GIII	*PMS2*	PMS2-	MSI-H (NR21,BAT26)
54	Johannesma [52]	1	11	F	Papillary glioneural tumor	*PMS2*	PMS2–	MSS(BAT26, BAT25, NR21, NR24, MONO 27, Penta D, Penta C)
55	Baas [8]	1	3	M	GBM	*MLH1*	MLH1–	NA
56		2	2	M	Astrocytoma of spinal cord	*PMS2*	§MLH1–/PMS2-	NA
57	Lindsay [53]	1	12	M	Medulloblastoma	*PMS2*	NA	NA
58	Walter [54]	1	13	F	GBM	*PMS2*	NA	NA
59	Yeung [55]	1	3	M	Optic pathway glioma	*PMS2*	NA	NA
60	Chmara [7]	1	11	M	Anaplastic oligodendroglioma	*PMS2*	NA	NA
61		2	4	M	GBM	*NA*	NA	NA
62		3	9	M	GBM	*PMS2*	NA	NA
63		4	4	F	GBM	*PMS2*	NA	NA
64	Bakry [56]	1	5	NS	Pleomorphic xantho-astrocytoma	*PMS2*	PMS2–	NS
65		2	8	NS	GBM	*MSH6*	MSH6–	NS
66		3	11	NS	GBM	*PMS2*	PMS2–	NS
67		4	13	NS	GBM	*PMS2*	PMS2–	NS
68		5	4	NS	GBM	*PMS2*	NA	NS
60		6	21	NS	Oligodendroglioma	*PMS2*	NA	NS
70		7	24	NS	GBM	*PMS2*	NA	NS
71		8	10	NS	Anaplastic oligodendroglioma	*MSH6*	MSH6–	NS
72		9	11	NS	Anaplastic astrocytoma	*MSH6*	MSH6–	NS
73		10	17	NS	GBM	*PMS2*	PMS2–	NS
74		11	12	NS	GBM	*MSH6*	NA	NS
75		12	8	NS	GBM	NA	PMS2–	NS
76		13	9	NS	GBM	NA	PMS2–	NS
77		14	4	NS	GBM	NA	PMS2–	NS
78		15	16	NS	Anaplastic astrocytoma	NA	PMS2–	NS
79		16	11	NS	Pleomorphic xantho-astrocytoma	NA	PMS2–	NS
80	Bougeard [57]	1	11	F	GBM	*PMS2*	* PMS2–	*MSI-H
81		2	9	M	GBM	*PMS2*	* PMS2–	*MSI-H
82	Daou [58]	1	22	F	Anaplastic ganglioglioma	*PMS2*	*PMS2–*	NA
83	Lavoine [59]	1	21	NS	Ganglioglioma	*PMS2*	PMS2–	MSI-H
84		2	40	NS	Glioblastoma	*PMS2*	NA	NA
85		3	11	NS	Glioblastoma	*PMS2*	# MLH1-/PMS2-	MSI-H
86		4	6	NS	GBM	*PMS2*	# MLH1-/PMS2-	MSS
87		5	13	NS	GBM	*PMS2*	PMS2–	NA
88		6	22	NS	GBM	*PMS2*	PMS2–	MSI-H
89		7	32	NS	GBM	*PMS2*	PMS2–	MSI-H
90		8	34	NS	GBM	*PMS2*	PMS2–	MSI-H
91		9	6	NS	Spinal GBM	*PMS2*	PMS2–	NA
92		10	6	NS	GBM	*PMS2*	NA	NA
93		11	5	NS	Medulloblastoma	*MLH1*	# MLH1–/PMS2–	MSS
94		12	5	NS	Oligodendroglioma	*MLH1*	# MLH1–/PMS2–	MSS
95		13	6	NS	GBM	*MSH6*	MSH6–	MSS
96		14	9	NS	Astrocytoma	*MSH6*	MSH6+	MSS
97		15	14	NS	GBM	*MSH6*	# MSH2–/MSH6–	MSS
98	Nguyen [60]	1	6	F	Medulloblastoma	*MLH1*	§ MLH1–/PMS2-	MSI-H (TP53, D17S250, D2S123, D5S346)
99		2	5	M	Glioma IV	*MLH1*	§ MLH1-/PMS2-	MSI-H (TP53, D17S250, D2S123,D5S2013)
100		3	5	M	Brain tumor	*NA*	NA	NA
101	Maletzki [61]	1	11	F	Anaplastic astrocytoma	*MSH6*	NA	MSI-H (BAT25, BAT26, Cat25, Bat40, NR24, MRPL2, TP53, DAMS)
102		2	10	M	GBM	*MSH6*	NA	MSI-H (BAT25, BAT26, Cat25, Bat40, NR24, TP53, DAMS)
103		3	4	F	GBM	*PMS2*	NA	MSI-H (BAT25, BAT26, Cat25, Bat40, NR21, NR24, NR27, MRPL2, TP53, DAMS)
104	Taeubner [62]	1	13	F	Medulloblastoma	*MSH6*	MSH6 +	NA
105	AlHarbi [63]	1	5	F	GBM	*MSH6*	NA	NA
106	Baig [64]	1	11	M	GBM	*PMS2*	NA	NA
107		2	9	F	PNET, astrocytoma	NA	NA	NA
108		3	5	M	PNET	NA	NA	NA
109		4	1	M	PNET	NA	NA	NA
110		5	7	M	GBM	*PMS2*	NA	NA
111		6	9	F	PNET, astrocytoma	*PMS2*	NA	NA
112	Bush [65]	1	27	F	GBM	*MSH6*	NA	NA
113	Farah [66]	1	10	M	High-grade Glioma	*PMS2*	PMS2–	NA
114		2	22	F	GBM	*PMS2*	PMS2–	NA
115		3	10	M	Medulloblastoma	*PMS2*	PMS2-	NA
116		4	12	F	Brain tumor	*PMS2*	PMS2-	NA
117	Current study	1	7	M	GBM	*MSH6*	MSH6-	MSS
118		2	4	M	Malignant astrocytoma	*MSH6*	MSH6-	MSI-L (NR27)

Abbreviations: IHC, immunohistochemistry; MSI, microsatellite instability; GBM, glioblastoma multiforme; NA, not analyzed; NS, not specified; MSI-H, microsatellite instability, high grade; MSI-L, microsatellite instability, low grade; MSS, microsatellite instability, stable; § MLH1, staining loss in tumor cells but not in non-neoplastic cells. # IHC, immunohistochemical pattern not specified; *, results from Lavoine et al., [59].

**Table 5 ijms-22-04629-t005:** Phenotypic features of patients who did not meet criteria for MMR IHC screening or had IHC false-negative results.

Case ID	Reference (Number)	Reason for Non-Selection by IHC Screening	Malignancies (Age at Diagnosis)	Others (Age at Diagnosis)	Points *
37	Senter [43]	Age at brain tumor diagnosis	Glioma (35) Rectum (24) Endometrial (35)	Brother: CRC (26), glioblastoma (34) Brother: Glioma (24)	5
46	Herkert [49]	Age at brain tumor diagnosis	GBM (34) CRC (21) Duodenal (32) Jejunal (×2) (34)	CR polyps (11) GI polyps with dysplasia (21 and 32), congenital asplenia, left isomerism, ventricle septum defect LS family	6
84	Levoine [59]	Age at brain tumor diagnosis	Glioblastoma (40)CRC (22 and 32)	Polyps (38) CALMs LS family	5
89	Levoine [59]	Age at brain tumor diagnosis	GBM (32) CRC (20)Gastric (32)	Polyps (20)	3
90	Levoine [59]	Age at brain tumor diagnosis	GBM (34) CRC (22 and 25) Endometrial (35) Small-bowel (36)	Polyps (22) Consanguinity LS family	4
112	Bush [65]	Age at brain tumor diagnosis	GBM (27)	Multiple polyps (23) CALMs Sister: 2 adenomas and 1 with high-grade dysplasia (21) breast cancer (29) LS family	3
42	Sjursen [46]	PMS2+	Giant cell glioblastoma (10) CRC (20) Duodenal (26) Ileal (30 and 36) Endometrial (31) Jejunal (42) Basal cell carcinoma several times	Polyps Sister: CRC (16)Father: Gastric cancer (64)	4
7	Menko [30]	MSH6+	Oligodendroglioma (10) CRC (12)	CALMs Consanguinity	8
96	Levoine [59]	MSH6+	Astrocytoma (9)	CALMs	4
104	Taeubner [62]	MSH6+	Medulloblastoma (13)	CALMs Other skin lesions reminiscent of NF1	4

Abbreviations: CR, colorectal; CRC, colorectal carcinoma; GI, gastrointestinal; GBM, glioblastoma multiforme; CALMs, *café-au-lait* macules, LS, Lynch syndrome; *, according to C4CMMRD clinical scoring system [3]; NF1, neurofibromatosis type 1.

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
