# Peer review of "The Challenge of Diagnosing Constitutional Mismatch Repair Deficiency Syndrome in Brain Malignancies from Young Individuals"

_ijms, 2021, doi:10.3390/ijms22094629_

Round 1
Reviewer 1 Report
Summary
The review article of Carrato et al. describes the methodology and challenges of diagnosing constitutional mismatch repair deficiency (CMMRD) in patients with paediatric brain tumours. They use a study of two cases from one family and a review of the literature to support their conclusions, covering important diagnostic considerations for clinicians, pathologists, and molecular biologists. Practitioners require such knowledge of the CMMRD phenotype and its diagnostic challenges so that cases, such as those described in the review, are not missed. The authors propose that IHC analysis of MMR protein expression should be used in all childhood and adolescent (age <25 years) brain tumours.
Broad comments
The review article of Carrato et al. is a thorough exploration of the nuances of diagnosing CMMRD in paediatric or adolescent brain tumour patients, which is well supported by evidence from the literature as well as their two cases. However, some content could be moved or revised to better frame their points (please see Comment 1 below for details).
The review article also proposes that all paediatric and adolescent brain tumour patients should be tested for MMR protein loss in the tumour and adjacent normal tissue using IHC to screen for CMMRD. Whilst I agree with this proposal, it is made without strong evidence that this would be effective in practice, and could be discussed in more detail to make a more convincing argument. To support this proposal, the authors could elaborate on the following points:
- The expected sensitivity of this screening method, which could be estimated from the literature summarised in Table 2.
- Sporadic MMR deficient brain tumours can also benefit from personalised treatment. This benefit is described by the authors in general, but not specifically for the sporadic MMR deficient tumours that would be identified in addition to CMMRD-related tumours.
- Why the age group of <25 years was chosen. I expect this is to focus screening on CMMRD, but a counter argument is that MMR deficiency testing should be used on all brain tumours irrespective of age to predict therapeutic response and screen for Lynch syndrome as well as CMMRD. I’m not aware of any large studies that have used unselected screening of brain tumours to estimate the frequency of MMR deficiency using IHC, but one study of 71 patients (adult and childhood) found that ~10% of glioblastoma were MMR deficient using IHC (Tepeoglu et al. 2019, PMID: 31529454), and another of 56 patients (age range not specified) found that ~4% of primary glioblastoma, and ~26% of paired relapsed glioblastoma, were MMR deficient using IHC (Indraccolo et al. 2019, PMID: 30514778). Also, in Table 2 there are several CMMRD patients with glioblastoma diagnosed at an age >25 years.
- How to manage false negatives due to the antigenic missense variants that can cause CMMRD.
- How this proposal fits within a clinical pathway with the existing C4CMMRD Consortium clinical scoring system, and the International BMMRD Consortium diagnostic criteria. I believe that the most appropriate strategy would be to screen for CMMRD using both a well-established molecular test of the tumour (such as IHC) and the C4CMMRD clinical scoring system to select patients for germline genetic testing who either have an MMR deficient tumour or score 3 or more points, with a final diagnosis being made according to the recent International BMMRD Consortium guidelines.
- The additional evidence that may be needed to support this proposal. For example, what is the expected frequency of CMMRD in paediatric or adolescent brain tumour patients? I don’t think any empirical estimates exist for this currently.
The review article is generally well-written but a check of English grammar and spelling is needed throughout. I’ve addressed some of the grammar and spelling in my specific comments.
Specific comments
Comment 1: Whilst generally well-written, sometimes the order of points in the article does not flow. In particular, the Introduction and Discussion would benefit from a revision of structure and content to streamline them and make them more readable. A few suggestions follow.
- Introduction: The description of various neoplastic and non-neoplastic features of CMMRD (lines 47-59) could be used to highlight the diagnostic challenge of such a broad phenotype, and how these features can be used to screen for CMMRD. For example, the text could read: “The pleiotropic CMMRD phenotype is not specific to the syndrome as these features overlap with other childhood tumor syndromes, in particular NF1. These features are used in a scoring system from the European C4CMMRD Consortium to raise clinical suspicion of CMMRD [3].” Critically, the authors should elaborate on the C4CMMRD scoring system and International BMMRD Consortium guidelines in the Introduction as these are critical for CMMRD diagnostics in current practice. A description of the International BMMRD Consortia diagnostic guidelines in the Introduction is particularly important in my opinion to inform the reader before they read the study aim and the Results; currently a description is only given in the Discussion.
- Introduction: The estimates of affected gene frequency among CMMRD patients “PMS2 is the most frequently mutated… …homozygous EPCAM patient has been described [13]” (lines 59-61) would be better placed as the second sentence of the Introduction so that these estimates follow the description of the genetic cause of the syndrome.
- Introduction: The second paragraph (lines 67-77), which describes the identification of CMMRD tumours through IHC testing of tumours to detect loss of MMR protein expression, would be better placed later in the Introduction, specifically being merged with paragraph 5 (lines 89-96), which has overlapping content. By merging these two paragraphs and placing them as the penultimate paragraph of the Introduction, the authors would better highlight that reflex MMR protein assessment of brain tumours by IHC could help to address the under-diagnosis of CMMRD.
- Introduction: The third paragraph (lines 78-81) would be better positioned alongside a description of the International BMMRD Consortium diagnostic criteria: These diagnostic criteria describe how to interpret germline genetic test results to make a diagnosis given that VUS and other uncertainties of genetic diagnosis are frequently observed in CMMRD diagnostics. The guidelines, like lines 78-81, also refer to functional assays that can be used as ancillary tests to inform the interpretation of genetic test results.
- Discussion: The first paragraph (lines 189-198) repeats information from the Introduction and so could be summarised in one sentence, such as: “The identification of individuals with CMMRD syndrome remains a challenge and a high degree of clinical awareness of its features and diagnostic pitfalls is required.”
- Discussion: If described in the Introduction instead, the details of the International BMMRD Consortium diagnostic guidelines (lines 213-216) could be removed or shortened.
- Discussion: The sixth paragraph (lines 276-283) repeats that a multidisciplinary approach is needed to identify CMMRD patients, which is also discussed in lines 199-202. The sixth paragraph also mentions the benefit of clinical identification of the syndrome to the patient. Therefore the sixth paragraph could be deleted to avoid repetition, and its content describing the benefit to CMMRD patients could be moved to follow lines 199-202.
Comment 2: The Abstract lines 27-28 could be clarified, for example: “CMMRD is underdiagnosed and often debuts with pediatric malignant brain tumors. A high degree of clinical awareness of the CMMRD phenotype is needed to identify new cases.”
Comment 3: Introduction lines 49-50 is grammatically incorrect, and could instead read: “Other LS-related neoplasms, such as endometrial and urinary tract carcinomas, can also develop.”
Comment 4: Introduction lines 97-100 is a long sentence and could be divided into two. It only mentions that the case studies were used, but the literature review and Table 2 are also important. The study aim could instead read: “The aim of this report is to draw attention to the peculiar IHC profile that characterizes the CMMRD syndrome, and to review the methods and difficulties in reaching a correct diagnosis. We demonstrate this by the case of two children with biallelic MSH6 germline mutations and brain tumors, and a review of the literature.”
Comment 5: Results lines 108-109 includes a typo, and should read: “The patient died at the age of 8 due to a spinal cord relapse of the tumor.”
Comment 6: Results lines 109-110 is missing a possessive on “brother”, and should read: “The younger brother was born with a single CALM but was considered as an NF1 patient due to his older brother’s medical history.”
Comment 7: Results lines 144-146 is not clear, and could instead read: “This may represent a germline variant as this specific marker is known to be quasi-monomorphic [69].” Also, reference 69 is cited but I think the authors mean to cite reference 70 (Bacher et al. 2004) or, more likely, reference 76 (Buhard et al. 2006) as this reference specifically deals with polymorphism frequency of each marker. Also, the identity of the microsatellite markers is not clear. Buhard et al. is referenced in methods (line 315) and uses BAT-25, BAT-26, NR-21, NR-22, and NR-24, but NR-22 is absent from this manuscript and NR-27 is used instead. I think this may be mixing up the markers from Buhard et al. with those used in the Promega MSI Analysis System, which uses a marker called MONO-27 instead of NR-22. Please clarify.
Comment 8: Results Figure 3 legend lines 149-150 need to be checked for grammar, and instead could read: “A broader spectrum is observed in marker NR-27 in the malignant astrocytoma.”
Comment 9: Results section “2.4. Somatic mutational analysis” (lines 161-168) describes the analysis of POLE and POLD1 genes in one of the brain tumours. I think this section needs restructuring so that the analysis is justified before the results are given, for example, the text could read: “Polymerase proofreading defects are characteristic of CMMRD brain tumors, and such tumors respond well to immune checkpoint blockade therapy due to their high mutation burden [66,71,72]. Therefore, we analysed POLE and POLD1 genes in the brain tumor DNA sample from patient III-2 and found 4 variants in POLD1 (Table 1).” It would be interesting to know what the mutation burden and signature of the tumour are, but these data are not critical.
Comment 10: Results lines 173-174 is not clear, and could instead read: “In order to compare our findings with those previously reported, the results of MMR protein expression using IHC and MSI analysis from a total of 119 reported brain tumors…”
Comment 11: Summary statistics from Table 2 are not described in the Result section, but are instead given in the Discussion (lines 236 -239). Also, these cover only the results from MSI analysis and not from IHC of MMR proteins. A summary of results for both IHC and MSI analysis should be described in the Results section with Table 2, i.e. what percentage of the tumours analysed by each method were MSI-H/MMR deficient.
Comment 12: The results from Amayari et al. are interesting but it’s not clear why these have been discussed separately in the Results (lines 181-187) and are not included in Table 2. Is it because the CMMRD diagnoses in these patients were not confirmed? If this is the case then I agree the data from Amayari et al. should not be included in Table 2. However, these data may be better placed in the Introduction as an example of how CMMRD may be under-diagnosed, particularly in populations with a high prevalence of consanguineous marriage, rather than in Results as these data alone are not a Result of this study.
Comment 13: The Discussion lines 199-202 could be clarified and simplified. Also, I assume “digestologist” should be “gastroenterologist”. For example, the text could read: “The cases presented here illustrate how, a multidisciplinary approach, with the participation of pathologists, dermatologists, geneticists, pediatricians, gastroenterologists, and neurooncologists, is needed to ensure an early diagnosis of CMMRD in brain tumor patients”
Comment 14: Discussion line 232 could instead read: “… PMS2 staining in tumor and normal tissue [7].”
Comment 15: Discussion lines 235-236 is not clear, and could instead read: “Another widely available assay to guide germline diagnosis of CMMRD is tumor MSI analysis, but CMMRD brain tumors were often MSS or MSI-L.”
Comment 16: Discussion line 245 refers to “dinucleotide markers”, but technically these should be referred to as “dinucleotide repeat markers”. There are several other instances where microsatellites are referred to as “mononucleotides” or “dinucleotides” when they should be “mononucleotide repeats” or “dinucleotide repeats” – please correct throughout the manuscript. Also, the authors could mention here that, although dinucleotide repeats have sometimes been shown to be better for the detection of increased MSI in brain tumours than mononucleotide repeats, they are insensitive to MSH6 deficiency.
Comment 17: Following the discussion of MSI analysis, the authors could also discuss that, whilst MSI analysis is ineffective and IHC can fail to detect MMR missense variants, sequencing of brain tumours is another option to screen for CMMRD. Sequencing can determine mutation burden and signatures, as well as detect the MMR and POLE or POLD1 variants, that are characteristic of CMMRD brain tumours.
Comment 18: In Material and methods the authors describe the use of internal positive controls for IHC assessment of tumour tissue (lines 306-307), but should mention that an independent tissue section was used as an additional positive control, given that internal positive control staining is not possible in CMMRD tissues.
Comment 19: In Conclusions line 346 there is a typo, the text should read: “…lack of family history of LS spectrum neoplasms.”
Author Response
Response to Reviewer 1 Comments
Comments and Suggestions for Authors
Summary
The review article of Carrato et al. describes the methodology and challenges of diagnosing constitutional mismatch repair deficiency (CMMRD) in patients with paediatric brain tumours. They use a study of two cases from one family and a review of the literature to support their conclusions, covering important diagnostic considerations for clinicians, pathologists, and molecular biologists. Practitioners require such knowledge of the CMMRD phenotype and its diagnostic challenges so that cases, such as those described in the review, are not missed. The authors propose that IHC analysis of MMR protein expression should be used in all childhood and adolescent (age <25 years) brain tumours.
Broad comments
The review article of Carrato et al. is a thorough exploration of the nuances of diagnosing CMMRD in paediatric or adolescent brain tumour patients, which is well supported by evidence from the literature as well as their two cases. However, some content could be moved or revised to better frame their points (please see Comment 1 below for details).
The review article also proposes that all paediatric and adolescent brain tumour patients should be tested for MMR protein loss in the tumour and adjacent normal tissue using IHC to screen for CMMRD. Whilst I agree with this proposal, it is made without strong evidence that this would be effective in practice, and could be discussed in more detail to make a more convincing argument. To support this proposal, the authors could elaborate on the following points:
The expected sensitivity of this screening method, which could be estimated from the literature summarised in Table 2.
ANSWER
We thank the reviewer for the comment.
Sensitivity of the MMR IHC was calculated and the result is on lines 239-241 as follows:
“The sensitivity of MMR IHC for identifying patients affected by CMMRD was 93.2%, as it showed loss of expression in at least one protein in 55 of the 59 brain neoplasms analyzed.”
Sporadic MMR deficient brain tumours can also benefit from personalised treatment. This benefit is described by the authors in general, but not specifically for the sporadic MMR deficient tumours that would be identified in addition to CMMRD-related tumours.
ANSWER
We thank the reviewer for the comment.
Microsatellite instability identifies neoplasms that benefit from immunotherapy treatment, mostly carcinomas.
According to brain malignancies and relating to prediction of therapeutic response, there are several works in the literature that postulated an “MMR-deficient” profile, as assessed by IHC in relapse tumoral tissue, as a mechanism of resistance of glioblastoma to TMZ, in patients who previously received TMZ. Different works define “IHC MMR-deficient” cases as those showing occasional negative cells (1) to more than 50% of tumoral negative cells (2), or even as a reduced immunostaining in tumor cells in comparison with internal normal control cells (3). These works, as well as other series, one of them published by our group (4), have explored MMR IHC loss in primary tumors (tissue obtained from the first surgery, previous to TMZ administration) with a very low frequency of negative cases. On the other hand, MMR IHC deficient pattern has also been explored in relation to pembrolizumab treatment in recurrent high-grade glioma with no relation between response and IHC pattern (5). The fact is that there is no relation between MMR IHC loss and MSI (2,5) and that the association between MMR status and hypermutation is still unclear in gliomas (5).
So, at this point, we think that MMR IHC analysis as a screening method to treatment prediction in all brain tumors is not justified.
- McCord M, Steffens A, Javier R, Kam KL, McCortney K, Horbinski C. The efficacy of DNA mismatch repair enzyme immunohistochemistry as a screening test for hypermutated gliomas. Acta Neuropathol Commun. 2020 Feb 12;8(1):15
- Rodríguez-Hernández I, Garcia JL, Santos-Briz A, Hernández-Laín A, González-Valero JM, Gómez-Moreta JA, Toldos-González O, Cruz JJ, Martin-Vallejo J, González-Sarmiento R. Integrated analysis of mismatch repair system in malignant astrocytomas. PLoS One. 2013 Sep 20;8(9):e76401
- Indraccolo S, Lombardi G, Fassan M, Pasqualini L, Giunco S, Marcato R, Gasparini A, Candiotto C, Nalio S, Fiduccia P, Fanelli GN, Pambuku A, Della Puppa A, D'Avella D, Bonaldi L, Gardiman MP, Bertorelle R, De Rossi A, Zagonel V. Genetic, Epigenetic, and Immunologic Profiling of MMR-Deficient Relapsed Glioblastoma. Clin Cancer Res. 2019 Mar 15;25(6):1828-1837
- Balana C, Vaz MA, Manuel Sepúlveda J, Mesia C, Del Barco S, Pineda E, Muñoz-Langa J, Estival A, de Las Peñas R, Fuster J, Gironés R, Navarro LM, Gil-Gil M, Alonso M, Herrero A, Peralta S, Olier C, Perez-Segura P, Covela M, Martinez-García M, Berrocal A, Gallego O, Luque R, Perez-Martín FJ, Esteve A, Munne N, Domenech M, Villa S, Sanz C, Carrato C. A phase II randomized, multicenter, open-label trial of continuing adjuvant temozolomide beyond 6 cycles in patients with glioblastoma (GEINO 14-01). Neuro Oncol. 2020 Dec 18;22(12):1851-1861
- Lombardi G, Barresi V, Indraccolo S, Simbolo M, Fassan M, Mandruzzato S, Simonelli M, Caccese M, Pizzi M, Fassina A, Padovan M, Masetto E, Gardiman MP, Bonavina MG, Caffo M, Persico P, Chioffi F, Denaro L, Dei Tos AP, Scarpa A, Zagonel V. Pembrolizumab Activity in Recurrent High-Grade Gliomas with Partial or Complete Loss of Mismatch Repair Protein Expression: A Monocentric, Observational and Prospective Pilot Study. Cancers (Basel). 2020 Aug 14;12(8):2283
As we are considering brain tumors in a syndromic context that typically affects children and adolescents we do not believe we should expand on the discussion of MMR-deficient adult brain tumors.
To refer this point we have added a new sentence in Discussion lines 354-357 as follows:
“However, given the controversy in considering "MMR-deficient" in adult brain tumors as a predictor of treatment [76-78] and because these neoplasms are not included in the LS tumor spectrum, we do not believe appropriate to consider universal screening by IHC in brain tumors in adults older than 25 years.”
With new references, 76 for Indraccolo et al. 2019,77 for Balana et al, 2020, and 78 for Lombardi et al. 2020.
Why the age group of <25 years was chosen. I expect this is to focus screening on CMMRD, but a counter argument is that MMR deficiency testing should be used on all brain tumours irrespective of age to predict therapeutic response and screen for Lynch syndrome as well as CMMRD. I’m not aware of any large studies that have used unselected screening of brain tumours to estimate the frequency of MMR deficiency using IHC, but one study of 71 patients (adult and childhood) found that ~10% of glioblastoma were MMR deficient using IHC (Tepeoglu et al. 2019, PMID: 31529454), and another of 56 patients (age range not specified) found that ~4% of primary glioblastoma, and ~26% of paired relapsed glioblastoma, were MMR deficient using IHC (Indraccolo et al. 2019, PMID: 30514778). Also, in Table 2 there are several CMMRD patients with glioblastoma diagnosed at an age >25 years.
ANSWER
We thank the reviewer for the comment.
We decided to choose the age group of <25 years for CMMRD screening for consistency with published consensus guidelines (Wimmer, 2014 and Aronson 2021), although brain neoplasms in patients with CMMRD can occur in those older than 25 years. In the review in Table 2 there are 6 (5%) patients with CMMRD and brain tumor older than 25 years (27, 32, 34 (x2), 35, 40), 4 due to PMS2 and one due to MSH6. These patients could be excluded from immunohistochemical screening for CMMRD due to the age of presentation of the brain tumor, but the set of clinical features that make up the phenotype they present would identify them as CMMRD with the existing C4CMMRD Consortium clinical scoring system.
To show these results, a new table (Table 5) has been added in Discussion accompanying the following text (line 370-374):
“Thus, patients excluded from MMR IHC screening due to presenting with brain tumor over the age of 25 years would be identified as CMMRD using the existing clinical scoring system of the C4CMMRD consortium based on the set of clinical features that make up their phenotype. This is true for all 6 patients over 25 years of age in the series reviewed (Table 4).”
Table 5. Phenotypic features of patients who did not meet criteria for MMR IHC screening or had IHC false-negative results
Case ID |
Reference (number) |
Reason for non- selection by IHC screening |
Malignancies (age at diagnosis) |
Others (age at diagnosis) |
Points* |
38 |
Senter [44] |
Age at brain tumor diagnosis |
Glioma (35) Rectum (24) Endometrial (35) |
Brother: CRC (26), glioblastoma (34) Brother: Glioma (24) |
5 |
47 |
Herkert [50] |
Age at brain tumor diagnosis |
GBM (34) CRC (21) Duodenal (32) Jejunal (x2) (34) |
CR polyps (11) GI polyps with dysplasia (21 and 32), congenital asplenia, left isomerism, ventricle septum defect LS family |
6 |
85 |
Levoine [ 60] |
Age at brain tumor diagnosis |
Glioblastoma (40) CRC (22 and 32) |
Polyps (38) CALMs LS family |
5 |
90 |
Levoine [60] |
Age at brain tumor diagnosis |
GBM (32) CRC (20) Gastric (32) |
Polyps (20) |
3 |
91 |
Levoine [60] |
Age at brain tumor diagnosis |
GBM (34) CRC (22 and 25) Endometrial (35) Small-bowel (36) |
Polyps (22) Consanguinity LS family |
4 |
113 |
Bush [66] |
Age at brain tumor diagnosis |
GBM (27) |
Multiple polyps (23) CALMs Sister: 2 adenomas and 1 with high-grade dysplasia (21), breast cancer (29) LS family |
3 |
43 |
Sjursen [47] |
PMS2+ |
Giant cell glioblastoma (10) CRC (20) Duodenal (26) Ileal (30 and 36) Endometrial (31) Jejunal (42) Basal cell carcinoma several times |
Polyps Sister: CRC (16) Father: Gastric cancer (64) |
4 |
7 |
Menko [30] |
MSH6+ |
Oligodendroglioma (10) CRC (12) |
CALMs Consanguinity |
8 |
97 |
Levoine [60] |
MSH6+ |
Astrocytoma (9) |
CALMs |
4 |
105 |
Taeubner [63] |
MSH6+ |
Medulloblastoma (13) |
CALMs Other skin lesions reminiscent of NF1 |
4 |
Abbreviations: CR, colorectal; CRC, colorectal carcinoma; CALMs, café-au-lait macules, LS, Lynch syndrome; *, according to C4CMMRD clinical scoring system [3]
Regarding screening for MMR deficiency to all brain tumors regardless of age to predict therapeutic response has been answered in the previous point.
For screening for Lynch syndrome it would not be appropriate, as brain tumors are not considered a neoplasm of the LS spectrum.
How to manage false negatives due to the antigenic missense variants that can cause CMMRD.
ANSWER
We thank the reviewer for the comment.
In fact, there were 4 patients in this situation, 1 with PMS2+ and 3 with MSH6+. With the strategy to combine both the screening IHC and the clinical score the 4 patients can be identify.
In Discussion we have added the following changes and have included these cases in Table 5:
Line 291-294:
“This is observed in 4 tumors in the series reviewed (cases 7, 43, 97, and 105) and could explain the lower correlation between the loss of protein expression with the mutated gene in brain tumors compared to what occurs in other types of neoplasms such as colon and endometrial carcinoma.”
Line 373-375:
“And in addition, those brain tumors retaining PMS2 or MSH6 expression were also identified applying the clinical scoring system (Table 5).”
Table 5. Phenotypic features of patients who did not meet criteria for MMR IHC screening or had IHC false-negative results
Case ID |
Reference (number) |
Reason for non- selection by IHC screening |
Malignancies (age at diagnosis) |
Others (age at diagnosis) |
Points* |
38 |
Senter [44] |
Age at brain tumor diagnosis |
Glioma (35) Rectum (24) Endometrial (35) |
Brother: CRC (26), glioblastoma (34) Brother: Glioma (24) |
5 |
47 |
Herkert [50] |
Age at brain tumor diagnosis |
GBM (34) CRC (21) Duodenal (32) Jejunal (x2) (34) |
CR polyps (11) GI polyps with dysplasia (21 and 32), congenital asplenia, left isomerism, ventricle septum defect LS family |
6 |
85 |
Levoine [ 60] |
Age at brain tumor diagnosis |
Glioblastoma (40) CRC (22 and 32) |
Polyps (38) CALMs LS family |
5 |
90 |
Levoine [60] |
Age at brain tumor diagnosis |
GBM (32) CRC (20) Gastric (32) |
Polyps (20) |
3 |
91 |
Levoine [60] |
Age at brain tumor diagnosis |
GBM (34) CRC (22 and 25) Endometrial (35) Small-bowel (36) |
Polyps (22) Consanguinity LS family |
4 |
113 |
Bush [66] |
Age at brain tumor diagnosis |
GBM (27) |
Multiple polyps (23) CALMs Sister: 2 adenomas and 1 with high-grade dysplasia (21), breast cancer (29) LS family |
3 |
43 |
Sjursen [47] |
PMS2+ |
Giant cell glioblastoma (10) CRC (20) Duodenal (26) Ileal (30 and 36) Endometrial (31) Jejunal (42) Basal cell carcinoma several times |
Polyps Sister: CRC (16) Father: Gastric cancer (64) |
4 |
7 |
Menko [30] |
MSH6+ |
Oligodendroglioma (10) CRC (12) |
CALMs Consanguinity |
8 |
97 |
Levoine [60] |
MSH6+ |
Astrocytoma (9) |
CALMs |
4 |
105 |
Taeubner [63] |
MSH6+ |
Medulloblastoma (13) |
CALMs Other skin lesions reminiscent of NF1 |
4 |
Abbreviations: CR, colorectal; CRC, colorectal carcinoma; CALMs, café-au-lait macules, LS, Lynch syndrome; *, according to C4CMMRD clinical scoring system [3]
How this proposal fits within a clinical pathway with the existing C4CMMRD Consortium clinical scoring system, and the International BMMRD Consortium diagnostic criteria. I believe that the most appropriate strategy would be to screen for CMMRD using both a well-established molecular test of the tumour (such as IHC) and the C4CMMRD clinical scoring system to select patients for germline genetic testing who either have an MMR deficient tumour or score 3 or more points, with a final diagnosis being made according to the recent International BMMRD Consortium guidelines.
The additional evidence that may be needed to support this proposal. For example, what is the expected frequency of CMMRD in paediatric or adolescent brain tumour patients? I don’t think any empirical estimates exist for this currently.
ANSWER
We thank the reviewer for the comment and the suggestion.
We agree that MMR IHC screening should be integrated with the established consensus, clinical and diagnostic guidelines.
The evidence supporting the proposal is that combining MMR IHC with the phenotypic characteristics scoring is possible to identify all patients screened in this study as CMMRD.
To support the proposal and evidence of efficacy, we have added the following changes in the Discussion at the beginning of the new paragraph and added a new table (Table 5) (line 364-370):
“To integrate MMR IHC screening with the established consensus clinical guidelines [3,4], we believe that the best strategy to follow would be to screen for CMMRD using both the MMR IHC in brain malignancy neoplasms in younger than 25 years of age and the C4CMMRD clinical scoring system [3] to select patients for germline genetic testing, those with an MMR-deficient tumor or with a score of 3 or more points, and that the final diagnosis would be made according to the recent guidelines of the International CMMRD Consortium [4]. Thus, patients excluded from MMR IHC screening due to presenting with brain tumor over the age of 25 years would be identified as CMMRD using the existing clinical scoring system of the C4CMMRD consortium based on the set of clinical features that make up their phenotype. This is true for all 6 patients over 25 years of age in the series reviewed (Table 4). And in addition, those brain tumors retaining PMS2 or MSH6 expression were also identified applying the clinical scoring system (Table 5).”
The review article is generally well-written but a check of English grammar and spelling is needed throughout. I’ve addressed some of the grammar and spelling in my specific comments.
Specific comments
Comment 1: Whilst generally well-written, sometimes the order of points in the article does not flow. In particular, the Introduction and Discussion would benefit from a revision of structure and content to streamline them and make them more readable. A few suggestions follow.
Introduction: The description of various neoplastic and non-neoplastic features of CMMRD (lines 47-59) could be used to highlight the diagnostic challenge of such a broad phenotype, and how these features can be used to screen for CMMRD. For example, the text could read:
“The pleiotropic CMMRD phenotype is not specific to the syndrome as these features overlap with other childhood tumor syndromes, in particular NF1. These features are used in a scoring system from the European C4CMMRD Consortium to raise clinical suspicion of CMMRD [3].”
Critically, the authors should elaborate on the C4CMMRD scoring system and International BMMRD Consortium guidelines in the Introduction as these are critical for CMMRD diagnostics in current practice. A description of the International BMMRD Consortia diagnostic guidelines in the Introduction is particularly important in my opinion to inform the reader before they read the study aim and the Results; currently a description is only given in the Discussion.
ANSWER
We thank the reviewer for this critical recommendation.
New text suggested has been included in the Introduction as follows (line 62-65):
“The pleiotropic CMMRD phenotype is not specific to the syndrome as these features overlap with other childhood tumor syndromes, in particular NF1. These features are used in a scoring system from the European C4CMMRD Consortium to raise clinical suspicion of CMMRD (Table 1) [3]”
And 2 new tables corresponding to the C4CMMRD scoring system (Table 1) and International CMMRD Consortium guidelines (Table 2) have been included in the Introduction:
Table 1. Scoring system to determine germline testing in eligibility for CMMRD
Indication for CMMRD testing in a cancer patient |
≥3 points |
Malignancies/premalignancies: one is mandatory; if more than one is present in the patient, add the points |
|
Carcinoma from the LS spectrum* at age <25 years |
3 points |
Multiple bowel adenomas at age <25 years and absence of APC/MUTYH mutation(s) or a single high-grade dysplasia adenoma at age <25 years |
3 points |
WHO grade III or IV glioma at age <25 years |
2 points |
NHL of T-cell lineage or sPNET at age <18 years |
2 points |
Any malignancy at age <18 years |
1 point |
Additional features: optional; if more than one of the following is present, add the points |
|
Clinical sign of NF1 and/or ≥2 hyperpigmented and/or hypopigmented skin alterations Ø>1 cm in the patient |
2 points |
Diagnosis of LS in a first-degree or second-degree relative |
2 points |
Carcinoma from LS spectrum* before the age of 60 in first-degree, second-degree, and third-degree relative |
1 point |
A sibling with carcinoma from the LS spectrum*, high-grade glioma, sPNET or NHL |
2 points |
A sibling with any type of childhood malignancy |
1 point |
Multiple pilomatricomas in the patient |
2 points |
One pilomatricoma in the patient |
1 point |
Agenesis of the corpus callosum or non-therapy-induced cavernoma in the patient |
1 point |
Consanguineous parents |
1 point |
Deficiency/reduced levels of IgG2/4 and/or IgA |
1 point |
*Colorectal, endometrial, small bowel, ureter, renal pelvis, biliary tract, stomach, bladder carcinoma.
Abbreviations: CMMRD, constitutional mismatch repair deficiency; LS, Lynch syndrome; NHL, non-Hodgkin’s lymphomas; PNET, supratentorial primitive neuroectodermal tumors.
Table 2. CMMRD diagnostic criteria
Criterion |
|
Germline result* PMS2, MSH6, MSH2, MLH1 |
Positive ancillary testing† |
Clinical phenotype |
||
Definitive diagnosis (strong evidence of CMMRD) |
1 |
Biallelic pathogenic variants (P/P)*, confirmed in trans.‡ |
Not required, unless unaffected >25 years then one required.† |
Not required if under age 25 (if no malignancy over age 25, ancillary testing required). |
||
2 |
Biallelic P/LP or LP/LP* variants, confirmed in trans‡ |
One required† for hallmark CMMRD.¶ Two required† for C4CMMRD criteria.** |
Hallmark CMMRD cancer diagnosis¶ or C4CMMRD criteria of 3 points** (then two ancillary tests required). |
|||
3 |
Heterozygous P or LP variant (±VUS* or likely benign variants). |
One required†. |
Hallmark CMMRD cancer diagnosis.¶ |
|||
4 |
No P or LP MMR variants (including VUS/ VUS).†† Or no testing available (ie, deceased proband). |
Two required†. |
Hallmark CMMRD cancer diagnosis.¶ |
|||
Likely diagnosis (moderate evidence of CMMRD) |
5 |
Biallelic P/LP* or LP/LP variants, confirmed in trans.§ |
Not required. |
C4CMMRD criteria of 3 points.** |
||
6 |
No P or LP MMR variants (including VUS/ VUS).†† Or no testing available (ie, deceased proband). |
One required.† |
Hallmark CMMRD cancer diagnosis.¶ |
|||
|
|
|||||
|
7 |
Heterozygous P or LP variant or no testing available (ie, deceased proband). |
Two required.† |
C4CMMRD criteria of 3 points.** Individuals aged <18 with NF1 features (ie, no malignancy or polyposis history). Malignancy under age 30 |
||
*Biallelic–impacts same gene on both parental alleles (ie, PMS2/PMS2); P, pathogenic (ACMG C5); LP, likely pathogenic (ACMG C4); VUS (ACMG C3). Multigene panel testing is recommended to investigate overlapping conditions. Consider phenotype of individual to rule out overlapping syndromes. All families should be assessed in a specialised centre for diagnosis.
†Ancillary testing does not include tumour mutation burden and signature at this time. Functional testing should be published with proven high sensitivity and specificity performed in an accredited (eg, CAP-inspected) laboratory authorised to give a clinically usable report. If discrepancy occurs among tests, multiple ancillary tests should be used to reach more conclusive decision.
‡In trans variants can be proven by testing parents, offspring or other relatives. If unavailable to confirm variants in trans, individual should fulfil criterion 3.
- If unavailable to confirm variants in trans, individual should fulfil criterion 6.
¶Hallmark CMMRD cancer: glioma or CNS embryonal tumours25 years, haematological cancer (excluding Hodgkin’s lymphoma) <18 years, GI adenocarcinoma <25 years, or >10 adenomatous GI polyps <18 years (after ruling out polyposis conditions).
**C4CMMRD criteria outlined in table 1.
††Consanguinity further supports a diagnosis of CMMRD due to a homozygous MMR gene mutation that is unidentifiable.
‡‡Individuals with two positive ancillary tests for CMMRD in the absence of the described phenotype can be assessed on a case-by-case basis, but these are atypical CMMRD cases and additional assessment is required to determine surveillance.
ACMG, American College of Medical Genetics; CAP, College of American Pathologist; C4CMMRD, European Consortium Care for CMMRD; CMMRD, constitutional mismatch repair deficiency syndrome; CNS, Central Nervous System; GI, gastrointestinal; MMR, mismatch repair; NF1, neurofibromatosis type 1.
Introduction: The estimates of affected gene frequency among CMMRD patients “PMS2 is the most frequently mutated… …homozygous EPCAM patient has been described [13]” (lines 59-61) would be better placed as the second sentence of the Introduction so that these estimates follow the description of the genetic cause of the syndrome.
ANSWER
The changed suggested has been introduced (line 45-47):
“PMS2 is the most frequently mutated gene accounting for 60% of CMMRD cases. The remaining 40% are due to MSH6 (20-30%), MLH1 and MSH2 (10-20%) pathogenic variants [3].”
Introduction: The second paragraph (lines 67-77), which describes the identification of CMMRD tumours through IHC testing of tumours to detect loss of MMR protein expression, would be better placed later in the Introduction, specifically being merged with paragraph 5 (lines 89-96), which has overlapping content. By merging these two paragraphs and placing them as the penultimate paragraph of the Introduction, the authors would better highlight that reflex MMR protein assessment of brain tumours by IHC could help to address the under-diagnosis of CMMRD.
ANSWER
We thank the reviewer for this recommendation.
The two paragraphs have been merged into new one (line 126-140) and placed as the penultimate of the Introduction as follows:
“CMMRD tumor identification requires the use of the tests applied to LS identification, namely immunohistochemical (IHC) assessment of the four MMR proteins expression and/or analysis of microsatellite instability (MSI) [15]. In CMMRD, however, these studies show some peculiar findings whose knowledge is mandatory to avoid underdiagnosis of this rare syndrome. Specifically, loss of MMR protein immunoreaction is not only seen in tumor cells but also in normal cells. Moreover, in CMMRD brain malignancies the standard MSI analysis fails to show instability [2,7,27-67] although the use of an extended Bethesda panel with 10 additional mononucleotide repeats has shown to detect subtle frameshifts [62]. Nowadays, the use of MMR IHC is becoming increasingly widespread mainly due to universal LS screening of colorectal and endometrial tumors [68,69]. Its use will increase with the adoption of the tissue diagnostic approach, a revolutionary paradigm shift in cancer treatment and drug development [70]. Therefore, a good knowledge of the immunohistochemical pattern characteristic of CMMRD is crucial to avoid pitfalls in its interpretation, given the great diversity of lesions and neoplasms that may be the first manifestation of this syndrome.”
Introduction: The third paragraph (lines 78-81) would be better positioned alongside a description of the International BMMRD Consortium diagnostic criteria: These diagnostic criteria describe how to interpret germline genetic test results to make a diagnosis given that VUS and other uncertainties of genetic diagnosis are frequently observed in CMMRD diagnostics. The guidelines, like lines 78-81, also refer to functional assays that can be used as ancillary tests to inform the interpretation of genetic test results.
ANSWER
We thank the reviewer for this critical recommendation.
The sentence has been moved to the beginning of the second paragraph (line 73-75) and new text has been included in relation to the diagnostic criteria described by Aronson et al and preceding Table 2 (line 75-83).
“The identification of pathogenic variants in both alleles of an MMR gene is required to confirm the diagnosis. However, the presence of PMS2 pseudogenes [14] and variants of unknown significance (VUS) makes it difficult. To improve the previous consensus established [15] a new diagnostic criterion for CMMRD elaborated by an international panel of experts have recently been described to unmask this rare syndrome (Table 2) [4]. They established four definitive criteria to ensure diagnosis of the syndrome with strong evidence and three probable criteria with moderate evidence based on the combination of a) the MMR germline results, b) ancillary testing including the immunohistochemistry of the 4 MMR genes on non-neoplastic tissue, germline microsatellite instability [16], ex vivo MSI plus methylation tolerance , in vitro repair assay and NGS detection [18], low-level MSI in tissue [19,20] and c) clinical manifestations.”
Discussion: The first paragraph (lines 189-198) repeats information from the Introduction and so could be summarised in one sentence, such as: “The identification of individuals with CMMRD syndrome remains a challenge and a high degree of clinical awareness of its features and diagnostic pitfalls is required.”
ANSWER
We thank the reviewer for the recommendation.
The lines 189-199 has been deleted and substituted by the sentence proposed as follows (line 252-253):
“The identification of individuals with CMMRD syndrome remains a challenge and a high degree of clinical awareness of its features and diagnostic pitfalls is required.”
Discussion: If described in the Introduction instead, the details of the International BMMRD Consortium diagnostic guidelines (lines 213-216) could be removed or shortened.
ANSWER
We thank the reviewer for the recommendation.
Lines 213-216 at the beginning of the third paragraph have been deleted and now read (line 272-294):
“IHC study of MMR is one of the ancillary tests included since in CMMRD at least one protein must be totally negative in normal tissue. For that reason, any unusual immunostaining pattern of MMR protein in neoplastic and normal cell should be a clue to guide the germ-line analysis. Herein, in the two new cases of CMMRD by biallelic MSH6 mutations with brain neoplasm, the IHC study of MMR proteins was crucial to arrive at the correct molecular diagnosis. The characteristic pattern of expression seen in CMMRD, namely lack of nuclear expression in both the tumor and the normal cells with MSH6 protein was crucial. To avoid the pitfall of interpreting the lack of staining in all cells as a failure of the technique it is important to include an external positive control from another individual in each determination. MMR IHC is a useful tool for identifying CMMRD patients with brain neoplasms with a sensitivity of 93% based on the cases reviewed in Table 4, showing loss of expression in at least one protein in 55 of the 59 tumors analyzed. However, unusual results of MMR protein expression have also been found. Tumors from patients with a homozygous MLH1 mutation displayed negative MLH1 staining only in neoplastic cells, with loss of PMS2 in both neoplastic and normal cells [61]. Baas et al described a spinal cord astrocytoma from a biallelic PMS2 mutation carrier with loss of MLH1 in tumor cells probably as a result of somatic alteration, with lack of PMS2 staining in tumor and normal tissue [8]. In addition, MSH6 and PMS2 could retain the expression in tumors in the case of biallelic germline missense mutations because of residual protein expression [29,46,59,62]. This is observed in 4 tumors in the series reviewed (cases 7, 43, 97, and 105) and could explain the lower correlation between the loss of protein expression with the mutated gene in brain tumors compared to what occurs in other types of neoplasms such as colon and endometrial carcinoma.”
Discussion: The sixth paragraph (lines 276-283) repeats that a multidisciplinary approach is needed to identify CMMRD patients, which is also discussed in lines 199-202. The sixth paragraph also mentions the benefit of clinical identification of the syndrome to the patient. Therefore, the sixth paragraph could be deleted to avoid repetition, and its content describing the benefit to CMMRD patients could be moved to follow lines 199-202.
ANSWER
We thank the reviewer for the recommendation.
The sixth paragraph has been deleted and the content describing the benefit of the treatment has been moved to the second paragraph, line 258-262, as follows:
“The cases presented here illustrate how, a multidisciplinary approach, with the participation of pathologists, dermatologists, geneticists, pediatricians, gastroenterologists, and neurooncologists, is needed to ensure an early diagnosis of CMMRD in brain tumor patients as well as to provide the best treatment. MMR deficient high-grade gliomas are resistant to standard treatments with combination of temozolomide and radiation but sensitive to immunotherapy. In addition, the overall survival of patients with CMMRD and malignant brain tumors is poor and worse than seen in the same population with no predisposition to hereditary cancer [26]. (…) “
Comment 2: The Abstract lines 27-28 could be clarified, for example: “CMMRD is underdiagnosed and often debuts with pediatric malignant brain tumors. A high degree of clinical awareness of the CMMRD phenotype is needed to identify new cases.”
ANSWER
We thank the reviewer for the recommendation.
The original sentence has been replaced by the reviewer's proposal (lines 27-28):
Abstract: Biallelic germline mismatch repair (MMR) genes (MLH1, MHS2, MHS6, and PMS2) mutations are an extremely rare event that causes constitutional mismatch repair deficiency (CMMRD) syndrome. CMMRD is underdiagnosed and often debuts with pediatric malignant brain tumors. A high degree of clinical awareness of the CMMRD phenotype is needed to identify new cases. (…)”
Comment 3: Introduction lines 49-50 is grammatically incorrect, and could instead read: “Other LS-related neoplasms, such as endometrial and urinary tract carcinomas, can also develop.”
ANSWER
We thank the reviewer for this important observation.
We have introduced the correct sentence (line 52-53):
“Other LS-related neoplasms, such as endometrial and urinary tract carcinomas, can also develop.”
Comment 4: Introduction lines 97-100 is a long sentence and could be divided into two. It only mentions that the case studies were used, but the literature review and Table 2 are also important. The study aim could instead read: “The aim of this report is to draw attention to the peculiar IHC profile that characterizes the CMMRD syndrome, and to review the methods and difficulties in reaching a correct diagnosis. We demonstrate this by the case of two children with biallelic MSH6 germline mutations and brain tumors, and a review of the literature.”
ANSWER
We thank the reviewer for this important observation.
The long sentence of lines 97-100 has been divided following the reviewer’s suggestions as follows (line 141-144):
“The aim of this report is to draw attention to the peculiar IHC profile that characterizes the CMMRD syndrome, and to review the methods and difficulties in reaching a correct diagnosis. We demonstrate this by the case of two children with biallelic MSH6 germline mutations and brain tumors, and a review of the literature.”
Comment 5: Results lines 108-109 includes a typo, and should read: “The patient died at the age of 8 due to a spinal cord relapse of the tumor.”
ANSWER
We thank the reviewer for his appreciation.
The typo (of) has been deleted. Now it reads (line 152-153):
“The patient died at the age of 8 due to a spinal cord relapse of the tumor”.
Comment 6: Results lines 109-110 is missing a possessive on “brother”, and should read: “The younger brother was born with a single CALM but was considered as an NF1 patient due to his older brother’s medical history.”
ANSWER
We thank the reviewer for his appreciation.
The possessive on “brother” has been added. Now it reads (line 153-154):
“The younger brother was born with a single CALM but was considered as an NF1 patient due to his older brother’s medical history”.
Comment 7: Results lines 144-146 is not clear, and could instead read: “This may represent a germline variant as this specific marker is known to be quasi-monomorphic [69].” Also, reference 69 is cited but I think the authors mean to cite reference 70 (Bacher et al. 2004) or, more likely, reference 76 (Buhard et al. 2006) as this reference specifically deals with polymorphism frequency of each marker. Also, the identity of the microsatellite markers is not clear. Buhard et al. is referenced in methods (line 315) and uses BAT-25, BAT-26, NR-21, NR-22, and NR-24, but NR-22 is absent from this manuscript and NR-27 is used instead. I think this may be mixing up the markers from Buhard et al. with those used in the Promega MSI Analysis System, which uses a marker called MONO-27 instead of NR-22. Please clarify.
ANSWER
We thank the reviewer for the comment.
Lines 144-146 has been replaced by the suggested sentence. Now it reads (line 189-190):
“2.2. MSI analysis
Analysis of MSI showed that none of the brain tumors were MSI-H. In the GBM case all markers showed an apparently stable profile (Figure 3), whereas in the malignant astrocytoma case only the NR27 marker showed a broader profile. This may represent a germline variant as this specific marker is known to be quasi-monomorphic [71]. “
After revision we can confirm that BAT25, BAT26, NR21, NR24, and NR27 are the markers described in Buhard et al., 2006. (now reference 71) an those described by Bacher et al. 2004 are the following: BAT-25, BAT-26, NR-21, NR-24 and MONO-27 (now reference 74).
To avoid the confusion in Discussion we have modified the sentence on lines 189-190 as follows:
“NR27 is one of the five markers of the quasi-monomorphic panel (NR21, NR27, NR24, BAT25 and BAT26) described by Burhard et al [71] and different from those validated in colorectal cancer due to its better specificity that allows avoiding normal cells as a control [74].”
Comment 8: Results Figure 3 legend lines 149-150 need to be checked for grammar, and instead could read: “A broader spectrum is observed in marker NR-27 in the malignant astrocytoma.”
ANSWER
We thank the reviewer for his appreciation.
The sentence has been corrected. Now it reads (line 193-194):
“A broader spectrum is observed in marker NR27 in the malignant astrocytoma.“
Comment 9: Results section “2.4. Somatic mutational analysis” (lines 161-168) describes the analysis of POLE and POLD1 genes in one of the brain tumours. I think this section needs restructuring so that the analysis is justified before the results are given, for example, the text could read: “Polymerase proofreading defects are characteristic of CMMRD brain tumors, and such tumors respond well to immune checkpoint blockade therapy due to their high mutation burden [66,71,72]. Therefore, we analysed POLE and POLD1 genes in the brain tumor DNA sample from patient III-2 and found 4 variants in POLD1 (Table 1).” It would be interesting to know what the mutation burden and signature of the tumour are, but these data are not critical.
ANSWER
We thank the reviewer for the comment and the suggestion.
Results section “2.4 Somatic mutational analysis” has been restructured in accordance with the reviewer’s suggestion (line 206-211):
“Acquired polymerase proofreading defects are characteristic of CMMRD brain tumors, leading to high mutational burden and associated with response to immune checkpoint inhibitors [66,71,72]. Therefore, the analysis of POLE and POLD1 genes in the brain tumor DNA sample from patient III-2 revealed the presence of 4 variants in POLD1 (Table 1). Of note, one of these variants, p.Glu318Lys affected the catalytic residue of POLD1 and is classified as pathogenic. Unfortunately, the mutational analysis was performed after the patient’s death and could not be used to guide an appropriated treatment.”
We agree with the reviewer that the information about tumor mutational burden and mutational signatures would be very interesting to obtain in order to fully characterize this type of childhood neoplasms. Unfortunately, the small size of the panel used (319 Kb) prevented these analyses.
A sentence has been added at the end of Methods section 4.4 in order to clarify this point (line 428-429):
“4.4. Targeted Next Generation Sequencing analysis
“The brain tumor DNA of patient III-2 was analyzed using a targeted Next Generation Sequencing custom panel, previously described in [83], that included the coding region of POLE and POLD1 genes among other genes (APC, AXIN2, BMPR1A; BUB1, BUB1B, BUB3, CDH1, CHEK2, ENG, EPCAM, EXO1, FAN1, MLH1, MLH3, MSH2, MSH3, MSH6, MUTYH, PMS1, PMS2, POLD1, POLE, PTEN, SMAD4, STK11 and TP53) and somatic hotspot mutations (design size: 319 Kb). Tumor from patient III-1 did not fulfill quality requirements to be analyzed. Briefly, capture of the target regions was performed using HaloPlex Target Enrichment kit 1-500 kb (Agilent Technologies, Santa Clara, CA, USA), according to the HaloPlex Target Enrichment System-Fast Protocol Version B. Library concentrations were normalized to 0.44 nM. Pooled libraries were sequenced in a MiSeq (Illumina, San Diego, CA, USA) with paired-end 250 bp reads plus an 8-base index read, using MiSeq Reagent Kit v3. Agilent SureCall application was used to trim, align and call variants. Variant filtering was performed based on Phred quality ≥30, alternative allele ratio ≥0.20, read depth ≥30x in the analyzed FFPE sample. Identified variants were then filtered against common single-nucleotide polymorphisms (MAF>1 according to the Exome Aggregation Consortium (ExAC) and the NHLBI GO Exome Sequencing Project (ESP) data). Pathogenicity assessment of the identified variants was aided by Alamut software v2.9.0. The size of the panel used prevented the analysis of tumor mutational burden or mutational signatures.”
Comment 10: Results lines 173-174 is not clear, and could instead read: “In order to compare our findings with those previously reported, the results of MMR protein expression using IHC and MSI analysis from a total of 119 reported brain tumors…”
ANSWER
We thank the reviewer for his appreciation.
The sentence has been modified. Now it reads (line 217-220):
“In order to compare our findings with those previously reported, the results of MMR protein expression using IHC and MSI analysis from a total of 119 reported brain tumors in patients with CMMRD syndrome have been reviewed and are specified in Table 4 [2,7,27-67].”
Comment 11: Summary statistics from Table 2 are not described in the Result section, but are instead given in the Discussion (lines 236 -239). Also, these cover only the results from MSI analysis and not from IHC of MMR proteins. A summary of results for both IHC and MSI analysis should be described in the Results section with Table 2, i.e. what percentage of the tumours analysed by each method were MSI-H/MMR deficient.
ANSWER
We thank the reviewer for this important recommendation.
Statistical data from Table 2 has been added as follows (line 227-250):
“Of the 119 patients in the series, the gender was specified in 78, being 43 (55%) males and 35 (45%) females.
Regarding the distribution of tumors by age, 73 (62%) appeared in the first decade of life, 26 (22%) between 11 and 15 years, 6 (5%) between 16 and 20, 7 (5%) between 21 and 24 and 6 (5%) over 25 years.
The most frequently altered gene was PMS2 with 68 (62%) cases, followed by MSH6 (28, 26%), MLH1 (10, 9%) and MHS2 (3, 3%). In 10 cases there is no data available on the altered gene.
MMR IHC was not performed in 60 tumors. Among the 59 cases analyzed, loss of PMS2 expression was the most frequent finding with 31 (52.5%), MSH6 was negative in 14 (23.7%), MLH1 and PMS2 negative in 7 (11.8%), MLH1 negative in 2 (3%), MSH2 and MSH6 negative in 1 (1.5%). PMS2 was positive in 1 (1.5%) (case 43) and MSH6 positive in 3 (5%) (cases 7, 97, and 105). The sensitivity of MMR IHC for identifying patients affected by CMMRD was 93.2%, as it showed loss of expression in at least one protein in 55 of the 59 brain neoplasms analyzed. In 54 cases it was possible to compare the result of germline analysis with MMR IHC and a good agreement was observed in 47 (87%). The remainder 7 (13%) with non-concordant results were as follow: 3 mutated MHS6 with MSH6 positive (cases 7, 97, and 105), 2 mutated PMS2 with MLH1 and PMS2 negative (cases 86 and 87), 1 mutated MSH6 with MSH2 and MSH6 negative (case 98) and 1 mutated PMS2 with PMS2 positive (case 43).
MSI analysis was performed in 56 (47%) tumors, but in 16 the results were not specified [49]. The results of the 40 tumors with available data showed that MSI-H was the most frequent finding with 19 (47.5%) followed by MSS with 16 (40%) and MSI-L with 5 (12.5%).”
Lines 236-236 in the Discussion section have been moved to Results (line 248-250):
“The results of the 40 tumors with available data showed that MSI-H was the most frequent finding with 19 (47.5%) followed by MSS with 16 (40%) and MSI-L with 5 (12.5%).”
Comment 12: The results from Amayari et al. are interesting but it’s not clear why these have been discussed separately in the Results (lines 181-187) and are not included in Table 2. Is it because the CMMRD diagnoses in these patients were not confirmed? If this is the case then I agree the data from Amayari et al. should not be included in Table 2. However, these data may be better placed in the Introduction as an example of how CMMRD may be under-diagnosed, particularly in populations with a high prevalence of consanguineous marriage, rather than in Results as these data alone are not a Result of this study.
ANSWER
We thank the reviewer for comments and recommendations.
The results of Amayari et al. were not included in Table 2 because the diagnoses of CMMRD were not confirmed.
Following the reviewer’s recommendation, the data from this study have been moved in the Introduction (line 116-125):
“CMMRD remains an underdiagnosed syndrome that usually debuts with malignant brain tumors in pediatric age with some clinical features that should raise suspicion of this syndrome [26] and especially in the context of high consanguinity as shown by the study of Amayari et al [11]. In this study, the prevalence of CMMRD was evaluated among patients with malignant brain tumors younger than 18 years with a high rate of consanguinity. Among the 36 high-grade gliomas, 17 (39%) showed a lack of staining for an MMR protein, and of these 82% of cases showed the characteristic pattern of CMMRD. Patients with lack of staining in both neoplastic and normal cells also had CALM, consanguinity, and a family history of cancer. In addition, the median age at diagnosis was 12.2 years, similar to that of sporadic tumors.”
Comment 13: The Discussion lines 199-202 could be clarified and simplified. Also, I assume “digestologist” should be “gastroenterologist”. For example, the text could read: “The cases presented here illustrate how, a multidisciplinary approach, with the participation of pathologists, dermatologists, geneticists, pediatricians, gastroenterologists, and neurooncologists, is needed to ensure an early diagnosis of CMMRD in brain tumor patients”
ANSWER
We thank the reviewer for his appreciation.
We have followed the reviewer's comment to improve the sentence. Now it reads (line 255-258):
“The cases presented here illustrate how, a multidisciplinary approach, with the participation of pathologists, dermatologists, geneticists, pediatricians, gastroenterologists, and neurooncologists, is needed to ensure an early diagnosis of CMMRD in brain tumor patients as well as to provide the best treatment.”
Comment 14: Discussion line 232 could instead read: “… PMS2 staining in tumor and normal tissue [7].”
ANSWER
We thank the reviewer for his appreciation.
Tissue has been added (line 286-288). Now it reads:
“Baas et al described a spinal cord astrocytoma from a biallelic PMS2 mutation carrier with loss of MLH1 in tumor cells probably as a result of somatic alteration, with lack of PMS2 staining in tumor and normal tissue [8].”
Comment 15: Discussion lines 235-236 is not clear, and could instead read: “Another widely available assay to guide germline diagnosis of CMMRD is tumor MSI analysis, but CMMRD brain tumors were often MSS or MSI-L.”
We thank the reviewer for his appreciation.
Lines 235-236 has been changed by the suggested sentence (line 295-296):
“Another widely available assay to guide germline diagnosis of CMMRD is tumor MSI analysis, but CMMRD brain tumors were often MSS or MSI-L.”
Comment 16: Discussion line 245 refers to “dinucleotide markers”, but technically these should be referred to as “dinucleotide repeat markers”. There are several other instances where microsatellites are referred to as “mononucleotides” or “dinucleotides” when they should be “mononucleotide repeats” or “dinucleotide repeats” – please correct throughout the manuscript. Also, the authors could mention here that, although dinucleotide repeats have sometimes been shown to be better for the detection of increased MSI in brain tumours than mononucleotide repeats, they are insensitive to MSH6 deficiency.
ANSWER
We thank the reviewer for the comment.
Following the reviewer’s recommendation “dinucleotide markers” has been changed by “dinucleotide repeat markers” on lines 303, 306, and 309; “mononucleotide” has been changed by “mononucleotide repeats” on lines 312, and “microsatellite” has been changed but “repeat marker” on line 307.
Now it reads:
“In contrast, dinucleotide repeat markers have been shown to be most useful in CMMRD-associated brain tumors, although they are insensitive to MSH6 deficiency. In fact, it was well illustrated by Nguyen et al. analyzing different tumors of CMMRD patients with mononucleotide and dinucleotide repeat markers [61]. One sarcoma and two brain tumors were MSI-H only in dinucleotide repeat markers while one colorectal cancer was unstable in all types of markers. Of the 19 MSI-H brain tumors, the markers used in the analysis were detailed in 10 cases and in 9 of them the markers were dinucleotide repeats [26,39,42,53,54]. Bakry et al described 16 brain tumors and only one was MSI-H but the markers used were not specified [57]. Another way to enhance the efficiency of MSI analysis has been to extend the number of non-coding mononucleotide repeat markers.”
A commentary about the dinucleotide repeat markers has been added on lines 302-304 as follows:
“In contrast, dinucleotide repeat markers have been shown to be most useful in CMMRD-associated brain tumors, although they are insensitive to MSH6 deficiency.”
Comment 17: Following the discussion of MSI analysis, the authors could also discuss that, whilst MSI analysis is ineffective and IHC can fail to detect MMR missense variants, sequencing of brain tumours is another option to screen for CMMRD. Sequencing can determine mutation burden and signatures, as well as detect the MMR and POLE or POLD1 variants, that are characteristic of CMMRD brain tumours.
ANSWER
We thank the reviewer for this important observation.
New sentences addressing this issue has been added in Discussion as follows (line 316-326) with a new reference:
“Another possible option that would allow the identification of CMMRD in brain tumors would be sequencing to determine MSI, mutation burden and signatures and other alterations characteristic of these tumors such as POLE or POLD1 variants. In fact, a recent report has revealed novel signatures that are uniquely attributed to mismatch repair deficiency by using exome- and genome-wide microsatellite instability analysis [75]. In this study different microsatellite (MS)-mutated loci, lack of recurrently mutated MS-loci and lack of long MS-indels has been identified as the main differences between childhood and adult MMR-deficient cancers. These differences may explain why conventional electrophoretic MSI assays, based on the detection of indels of >3 bases in a small number of MS analyzed, were unable to detect MSI in pediatric tumors.”
- Chung. J.; Maruvka, Y.E.; Sudhaman, S.; Kelly J, Haradhvala, N.J.; Bianchi, V.; Edwards, M.; Forster, V.J.; Nunes, N.M.; Galati, M.A.; et al. DNA Polymerase and Mismatch Repair Exert Distinct Microsatellite Instability Signatures in Normal and Malignant Human Cells. Cancer Discov. 2020, Dec 18. doi: 10.1158/2159-8290.CD-20-0790.
Comment 18: In Material and methods the authors describe the use of internal positive controls for IHC assessment of tumour tissue (lines 306-307), but should mention that an independent tissue section was used as an additional positive control, given that internal positive control staining is not possible in CMMRD tissues.
ANSWER
We thank the reviewer for this important observation.
A new sentence has been added as follows (line 391-393):
“A separate tissue section was used as an additional positive control since internal positive control staining is not observed in CMMRD tissues.”
Comment 19: In Conclusions line 346 there is a typo, the text should read: “…lack of family history of LS spectrum neoplasms.”
ANSWER
We thank the reviewer for his appreciation.
The typo at line 437 has been corrected. Now it reads:
“Therefore, we propose the use of MMR IHC in malignant brain tumors diagnosed below 25 years of age to unmask those suffering from CMMRD that may have gone undetected due to the few key manifestations or the lack of family history of LS spectrum neoplasms.”

Reviewer 2 Report
This paper discusses the diagnostic challenge of constitutional DNA mismatch repair deficiency (CMMRD) behind childhood brain malignancies. A case report of two siblings with biallelic germline mutations of MSH6 is provided, together with a summary of mismatch repair (MMR) protein expression and microsatellite instability (MSI) findings from 119 published brain tumor cases from patients with CMMRD. Recognition of CMMRD is important for tailored therapy. The authors suggest immunohistochemical analysis of MMR proteins in all malignant brain tumors diagnosed below 25 years of age for early detection of CMMRD; lack of protein expression in both neoplastic and non-neoplastic cells would point to CMMRD. In contrast, MSI analysis is less useful, since approximately half of all reported brain tumors are microsatellite-stable or show only low degree of MSI by conventional techniques.
General comment: This is a useful review of CMMRD as an underlying cause of brain malignancies.
Specific comments:
- A somatic mutational analysis by panel sequencing was performed on a brain tumor from patient III-2, which resulted in the discovery of four POLD1 variants. Did the panel include other genes that would be relevant for the clinical and/or tumor phenotype; for example, were secondary somatic mutations in MMR genes observed? Was the overall number of somatic mutations in the hypermutated range (over 10 somatic mutations per megabase)?
- In Discussion, could the authors speculate on the possible mechanisms behind stable microsatellites in brain tumors? The finding is unexpected considering constitutional inactivation of both alleles of a given MMR gene in CMMRD patients.
Author Response
Response to Reviewer 2 Comments
Comments and Suggestions for Authors
This paper discusses the diagnostic challenge of constitutional DNA mismatch repair deficiency (CMMRD) behind childhood brain malignancies. A case report of two siblings with biallelic germline mutations of MSH6 is provided, together with a summary of mismatch repair (MMR) protein expression and microsatellite instability (MSI) findings from 119 published brain tumor cases from patients with CMMRD. Recognition of CMMRD is important for tailored therapy. The authors suggest immunohistochemical analysis of MMR proteins in all malignant brain tumors diagnosed below 25 years of age for early detection of CMMRD; lack of protein expression in both neoplastic and non-neoplastic cells would point to CMMRD. In contrast, MSI analysis is less useful, since approximately half of all reported brain tumors are microsatellite-stable or show only low degree of MSI by conventional techniques.
General comment: This is a useful review of CMMRD as an underlying cause of brain malignancies.
Specific comments:
A somatic mutational analysis by panel sequencing was performed on a brain tumor from patient III-2, which resulted in the discovery of four POLD1 variants. Did the panel include other genes that would be relevant for the clinical and/or tumor phenotype; for example, were secondary somatic mutations in MMR genes observed? Was the overall number of somatic mutations in the hypermutated range (over 10 somatic mutations per megabase)?
ANSWER
We thank the reviewer for the comment.
As detailed in Vargas-Parra et al. 2017 [ref. 78], our target panel was composed of DNA capture probes of 509 target regions, including the coding exons and promoter regions of 26 colorectal cancer (CRC) associated genes: APC, AXIN2, BMPR1A, BUB1, BUB1B, BUB3, CDH1, CHEK2, ENG, EPCAM, EXO1, FAN1, MLH1, MLH3, MSH2, MSH3, MSH6, MUTYH, PMS1, PMS2, POLD1, POLE, PTEN, SMAD4, STK11 and TP53. Additionally, somatic hotspot mutations in target genes (AKT1, BRAF, CTNNB1, EGFR, FBXW7, GNAS, KRAS, MAP2K1 (MEK1), MET, NRAS, PIK3CA and SRC) and MSI CRC‐associated loci of SETD2, SETD1B and SETDB2 were also included. Final design was 319,653 kb size.
The complete list of variants identified in the tumor analysis was not provided in the manuscript. The main reasons are the following: 1) a definitive identification of somatic variants was not possible since normal tissue DNA was not available in the deceased patient III-2; 2) the analyzed genes were not selected based on their relevance in brain tumors; and 3) the sequencing coverage of the FFPE sample analyzed was heterogeneous. As mentioned above (comment 9, reviewer 1), we agree with the reviewer that the confirmation of tumor mutational burden in the hypermutated range (over 10 somatic mutations per Mb) would be very interesting to obtain in order to fully characterize this type of childhood neoplasms. Unfortunately, the small size of the panel used (319 Kb) prevented this analysis.
Methods section “4.4. Targeted Next Generation Sequencing analysis” has been modified to clarify this issue.
4.4. Targeted Next Generation Sequencing analysis
The brain tumor DNA of patient III-2 was analyzed using a targeted Next Generation Sequencing custom panel, previously described in [83], that included the coding region of POLE and POLD1 genes among other genes (APC, AXIN2, BMPR1A; BUB1, BUB1B, BUB3, CDH1, CHEK2, ENG, EPCAM, EXO1, FAN1, MLH1, MLH3, MSH2, MSH3, MSH6, MUTYH, PMS1, PMS2, POLD1, POLE, PTEN, SMAD4, STK11 and TP53) and somatic hotspot mutations (design size: 319 Kb). Tumor from patient III-1 did not fulfill quality requirements to be analyzed. Briefly, capture of the target regions was performed using HaloPlex Target Enrichment kit 1-500 kb (Agilent Technologies, Santa Clara, CA, USA), according to the HaloPlex Target Enrichment System-Fast Protocol Version B. Library concentrations were normalized to 0.44 nM. Pooled libraries were sequenced in a MiSeq (Illumina, San Diego, CA, USA) with paired-end 250 bp reads plus an 8-base index read, using MiSeq Reagent Kit v3. Agilent SureCall application was used to trim, align and call variants. Variant filtering was performed based on Phred quality ≥30, alternative allele ratio ≥0.20, read depth ≥30x in the analyzed FFPE sample. Identified variants were then filtered against common single-nucleotide polymorphisms (MAF>1 according to the Exome Aggregation Consortium (ExAC) and the NHLBI GO Exome Sequencing Project (ESP) data). Pathogenicity assessment of the identified variants was aided by Alamut software v2.9.0. The size of the panel used prevented the analysis of tumor mutational burden or mutational signatures.
In Discussion, could the authors speculate on the possible mechanisms behind stable microsatellites in brain tumors? The finding is unexpected considering constitutional inactivation of both alleles of a given MMR gene in CMMRD patients.
ANSWER
We thank the reviewer for the comment.
A recent report has revealed novel signatures that are uniquely attributed to mismatch repair and DNA polymerase by using exome- and genome-wide microsatellite instability analysis (Chung et al., 2020). In this study different microsatellite (MS)-mutated loci, lack of recurrently mutated MS-loci and lack of long MS-indels has been identified as the main differences between childhood and adult MMR-deficient cancers. These differences may explain why conventional electrophoretic MSI assays, based on the detection of indels of >3 bases in a small number of MS analyzed, were unable to detect MSI in pediatric tumors.
Discussion section has been modified to include this hypothesis on lines 319-326 with a new reference:
“In fact, a recent report has revealed novel signatures that are uniquely attributed to mismatch repair deficiency by using exome- and genome-wide microsatellite instability analysis [75]. In this study different microsatellite (MS)-mutated loci, lack of recurrently mutated MS-loci and lack of long MS-indels has been identified as the main differences between childhood and adult MMR-deficient cancers. These differences may explain why conventional electrophoretic MSI assays, based on the detection of indels of >3 bases in a small number of MS analyzed, were unable to detect MSI in pediatric tumors.”
- Chung. J.; Maruvka, Y.E.; Sudhaman, S.; Kelly J, Haradhvala, N.J.; Bianchi, V.; Edwards, M.; Forster, V.J.; Nunes, N.M.; Galati, M.A.; et al. DNA Polymerase and Mismatch Repair Exert Distinct Microsatellite Instability Signatures in Normal and Malignant Human Cells. Cancer Discov. 2020, Dec 18. doi: 10.1158/2159-8290.CD-20-0790.

Round 2
Reviewer 1 Report
I thank the authors for their comprehensive response, my comments and suggestions have all been addressed. In particular, the proposal to use MMR IHC in CNS tumours of patients aged <25 years to screen for CMMRD is now well justified, and the manuscript structure and content has been significantly improved in my opinion.
I have two additional minor comments:
- In line 334 (when tracked changes are shown) I think there is a typo and that the sentence is meant to read: “The finding of a somatic pathogenic variant in POLD1…”
- In line 374 (when tracked changes are shown) there is a typo: “Burhard”
I believe this will be a very useful review for practitioners and scientists in this field.